# CDCP1 promotes compensatory renal growth by integrating Src and Met signaling

Kentaro Kajiwara[1] , Shotaro Yamano[2] , Kazuhiro Aoki[3], Daisuke Okuzaki[4] , Kunio Matsumoto[5], Masato Okada[1]

**Compensatory growth of organs after loss of their mass and/or function is controlled by hepatocyte growth factor (HGF), but the underlying regulatory mechanisms remain elusive. Here, we show that CUB domain-containing protein 1 (CDCP1) promotes HGF-induced compensatory renal growth. Using canine kidney cells as a model of renal tubules, we found that HGF-induced temporal up-regulation of Src activity and its scaffold protein, CDCP1, and that the ablation of CDCP1 robustly abrogated HGF-induced phenotypic changes, such as morphological changes and cell growth/proliferation. Mechanistic analyses revealed that up-regulated CDCP1 recruits Src into lipid rafts to activate STAT3 associated with the HGF receptor Met, and activated STAT3 induces the expression of matrix metalloproteinases and mitogenic factors. After unilateral nephrectomy in mice, the Met-STAT3 signaling is transiently up-regulated in the renal tubules of the remaining kidney, whereas CDCP1 ablation attenuates regenerative signaling and significantly suppresses compensatory growth. These findings demonstrate that CDCP1 plays a crucial role in controlling compensatory renal growth by focally and temporally integrating Src and Met signaling.**

## Introduction

Controlling organ size during development and/or regenerative growth is important for maintaining organ function, body homeostasis, and health. The kidneys are paired organs that generate urine through the filtration of blood and reabsorption of water and nutrients, and kidney mass is strictly correlated with total body mass. The renal tubules constitute most of the mass and function, and have a remarkable capacity to undergo regenerative growth. Unilateral nephrectomy (UNX), a surgical procedure to reduce kidney mass, increases fluid flow in the remaining kidney, and promotes the subsequent growth/hypertrophy and proliferation/hyperplasia of tubular epithelial cells to compensate for the increased

flow (1, 2). This compensatory renal growth is regulated by the activation of mechanistic target of rapamycin (mTOR) signaling pathways (3, 4, 5). However, interfering with the function of this pathway does not completely suppress renal growth, suggesting the potential contribution of one or more additional signaling pathways.

Compensatory renal growth also requires a number of growth factors (6), among which hepatocyte growth factor (HGF) plays an important role (7). HGF is produced by the surrounding or distal mesenchyme in the remaining kidney immediately after UNX (8, 9, 10), inducing the up-regulation of its receptor, Met, in the renal tubules (9). In addition, HGF promotes dynamic morphogenesis through the induction of epithelial–mesenchymal transition in epithelial cells during development and regenerative growth of the kidney (11, 12). HGF-mediated morphogenesis requires STAT3 signaling (13), which is regulated by Met through endosomal trafficking (14). However, the molecular mechanisms through which the multifaceted HGF functions are precisely controlled during compensatory renal growth remain elusive.

The MDCK cell line was derived from renal tubule epithelial cells and is a physiologically relevant in vitro model to study the regulation of HGF functions in the kidney (15, 16). When grown in three-dimensional cultures, MDCK cells spontaneously form spherical cysts that resemble renal tubules, comprising an epithelial monolayer and lumen. Upon stimulation with HGF, MDCK cysts undergo morphological alterations and form branched tubular structures (17, 18). During this morphogenesis, the MDCK cells lose their epithelial polarity via a partial epithelial–mesenchymal transition-like phenotypic change and protrude into the ECM by penetrating the basement membrane (19, 20). In addition, HGF promotes cell growth and proliferation, resulting in the formation of multi-cell layered cysts. To elucidate the mechanisms underlying HGF-induced phenotypic changes, the roles of multiple signaling axes downstream of Met, such as the Ras-ERK, Akt-mTOR, Src, and STAT3 pathways, have been investigated extensively (19, 20, 21, 22, 23). However, the molecular mechanism by which these diverse signaling pathways are accurately coordinated by HGF-Met needs to be clarified. Furthermore, the most important pathway for HGF-induced compensatory renal growth remains undefined.

[1]Department of Oncogene Research, Research Institute for Microbial Diseases, Osaka University, Osaka, Japan    [2]Japan Bioassay Research Center, Japan Organization of Occupational Health and Safety, Kanagawa, Japan    [3]Division of Quantitative Biology, Okazaki Institute for Integrative Bioscience, National Institute for Basic Biology, National Institutes of Natural Sciences, Aichi, Japan    [4]Genome Information Research Center, Research Institute for Microbial Diseases, Osaka University, Osaka, Japan    [5]Division of Tumor Dynamics and Regulation, Cancer Research Institute, Kanazawa University, Kanazawa, Japan

Correspondence: okadam@biken.osaka-u.ac.jp; kajiwara@biken.osaka-u.ac.jp

Here, using three-dimensional cultures of MDCK cysts as a structural model of renal tubules, we identified Src and its membrane scaffold protein, (CUB) complement C1r/C1s, Uegf, Bmp1 domain-containing protein 1 (CDCP1) (also known as Trask or CD318), as regulatory elements of HGF signaling. We then investigated the mechanisms underlying the functions of the CDCP1-Src axis in MDCK cysts and verified its physiological roles in compensatory renal growth using *Cdcp1*-knockout mice. The results obtained in this study demonstrates that CDCP1 plays a significant role in controlling HGF-induced compensatory renal growth by focally and temporally integrating Src and Met-STAT3 signaling in lipid rafts.

# Results

### CDCP1 is required for HGF signaling in MDCK cysts

MDCK cells cultured in a collagen matrix formed cyst structures with luminal space (Fig 1A). HGF treatment immediately promoted cell growth and proliferation (as indicated by increased cyst diameter and a high ratio of Ki67-positive cells) and morphological changes with some cysts exhibiting extended cell protrusions (15, 24) (Fig 1A–D). Similar HGF-induced effects were observed in primary cultured renal proximal tubule cells (25), indicating that MDCK cysts are suitable for the mechanistic analysis of HGF functions. To dissect the HGF-related signaling pathways, we first examined the effects of various signaling inhibitors on the HGF-induced phenomena. Treatment of MDCK cysts with NK4, a specific HGF antagonist (26), or Met kinase inhibitors robustly inhibited cell proliferation and morphological changes (Figs 1A–D and S1A–C), confirming that the observed phenomena were dependent on HGF. Treatment of the cysts with Torin1 or rapamycin, the selective mTOR inhibitors, suppressed cell growth/proliferation, but was less effective against the formation of cell protrusions (Fig 1A–D). In contrast, treatment with dasatinib or saracatinib, Src kinase inhibitors, potently suppressed both cell proliferation and protrusion extension (Fig 1A–D). Furthermore, immunofluorescence analysis revealed that a fraction of activated Src (pY418) was concentrated at the tip of the protruding cells (Fig 1E), which was visualized with mCherry-GPI, a lipid raft marker protein (Fig 1F). This phenomenon was validated using detergent-resistant membrane (DRM) separation analysis (Fig S1D). Alteration of lipid raft integrity by treatment with simvastatin, an inhibitor of cholesterol synthesis, reduced the HGF-induced phenomena (Fig 1A–D). These observations imply that HGF signaling is associated with the activation of Src in the lipid rafts in protruding cells.

To further assess its role in HGF signaling, we examined the effects of Src activation in MDCK cells by expressing Src-MER, a Src protein fused to a modified ligand-binding domain of the estrogen receptor (MER) (27) that could be activated by treatment with 4-hydroxytamoxifen (4-OHT) (Fig S2A and B). Src activation induced the formation of multiple cell protrusions and cell proliferation (Fig S2C), supporting the involvement of Src activity in HGF signaling. DRM separation analysis revealed that activated Src-MER was concentrated in the lipid raft fractions (Fig S2D). Because Src alone

can only partially localize to lipid rafts (28), we searched for scaffolding proteins that interact with and activate Src in these structures. Src-MER–interacting proteins were isolated from DRM fractions by co-immunoprecipitation and identified by mass spectrometry (Fig S2E and F). Among the candidate proteins identified, we selected the transmembrane glycoprotein, CDCP1, for further analysis because it is expressed in mouse renal proximal tubules (29), has palmitoylation sites required for lipid raft localization, and serves as a membrane adaptor of Src (30, 31) (Fig S3A). We then validated the role of CDCP1 in HGF signaling. Immunofluorescence analysis revealed that endogenous CDCP1 was transiently concentrated at the tips of protruding cells during the early stage of morphological changes (Fig 1G), in a manner similar to that of activated Src and mCherry-GPI (Figs 1E and F and S1D). More importantly, HGF-induced cell protrusion extension and cell proliferation were efficiently abrogated by *CDCP1*-knockout in MDCK cells (Figs 1H–J and S3B–D). These results suggest that the CDCP1-Src axis is a regulatory component of HGF signaling.

### Up-regulation of CDCP1 induces HGF-related phenotypic changes by activating Src in lipid rafts

To elucidate the mechanisms through which CDCP1 regulates HGF signaling, we generated MDCK cells in which the expression of exogenous CDCP1-EGFP could be controlled by doxycycline (Dox). Dox treatment induced the expression of full-length and cleaved CDCP1 at the plasma membrane (Fig S4A and B). DRM separation analysis showed that a fraction of CDCP1 was distributed in lipid raft-enriched fractions (Fig S4C). Moreover, induction of CDCP1 expression markedly increased the amount of phosphorylated Src (pY418) without affecting Src protein levels (Fig S4A), indicating that Src was activated by the up-regulation of CDCP1. Furthermore, activated Src was concentrated in the lipid raft fractions and physically associated with CDCP1 (Fig S4C and D). Multiple Src family kinases (SFKs) were expressed in MDCK cells, and Lyn and Yes also localized in lipid raft fractions (Fig S4E and F). After CDCP1 overexpression, Lyn and Yes relocated into the lipid raft fraction (Fig S4F and G), where activated Lyn and Yes associated with CDCP1 (Fig S4D). These results indicate that CDCP1 can associate with multiple activated SFKs. In this study, however, we hereafter focused our analysis on Src as a representative SFK. In contrast, a CDCP1 mutant lacking the Src-binding site (Y734F; CDCP1-YF) failed to activate Src (Fig S4A and C). A different CDCP1 mutant lacking the lipid raft localization signal (C689G–C690G; CDCP1-CG) was able to activate SFKs, but only in non-lipid raft fractions (Fig S4A and C). These findings corroborate the notion that CDCP1 specifically sequesters activated Src in lipid rafts.

Next, we examined the effects of CDCP1 expression on HGF-related phenotypes of MDCK cysts (Fig 2A). CDCP1-EGFP overexpression was induced after the completion of cystogenesis, and morphological changes were observed without HGF stimulation (Video 1). In the early stages (12–24 h after induction), the cells expressing CDCP1 gradually protruded toward the ECM. During the later stages (48 h after induction), multiple cell protrusions formed randomly and extended aggressively toward the ECM, with the formation of multi-layered cysts (Fig 2B). Concurrent with these phenomena, up-regulation of CDCP1 activated Src in the protruding

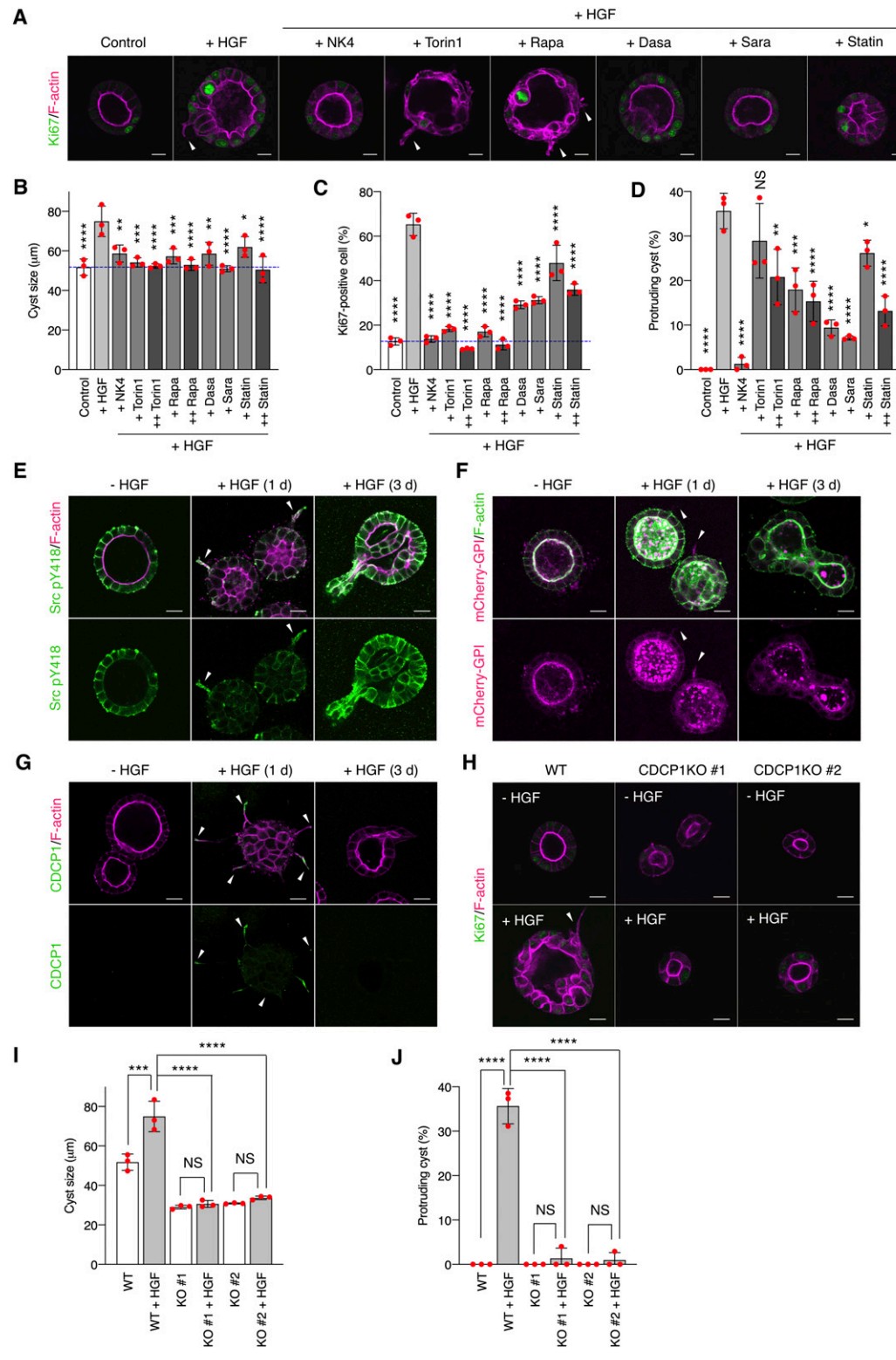

**Figure 1. CDCP1 is required for hepatocyte growth factor (HGF)–induced phenotypic changes.**
(A, B, C, D) MDCK cysts embedded within the collagen matrix were pretreated with NK4 (1 µg/ml), Torin1 (+, 50 nM; ++, 100 nM), rapamycin (+ Rapa, 50 nM; ++ Rapa, 100 nM), dasatinib (20 nM, Dasa), saracatinib (10 µM, Sara), or simvastatin (2 µM, Statin) and then incubated in the presence of HGF (50 ng/ml) for 1 d. **(A)** Ki67 was visualized with an Alexa Fluor 488–conjugated antibody (green), and actin filaments were stained with Alexa Fluor 594–phalloidin (magenta); arrowheads indicate transiently formed protrusions. Diameter (µm) of cysts (n = 100). **(B)** Dotted blue line indicates the average diameter of non-treated cysts. **(C)** Ratio of Ki67-positive cell in cyst (n = 50). **(D)** Fraction of the total number of cysts counted (n > 100) with protrusions. **(E)** MDCK cysts embedded within the collagen matrix were incubated in the presence of HGF

cells (Fig 2C), and inhibition of Src with dasatinib or saracatinib strongly suppressed CDCP1-induced cell proliferation and protrusion extension (Fig 2D and E and Table S1). Meanwhile, the up-regulation of CDCP1-YF or CDCP1-CG did not induce any significant phenotypic change (Fig 2F and G). Furthermore, CDCP1-induced protrusion extension was inhibited by depletion of membrane cholesterol (Fig 2H). Based on these results, it is likely that up-regulation of CDCP1 induces HGF-related phenotypic changes, that is, cell protrusion formation and cell proliferation, in MDCK cysts via the activation of Src in lipid rafts.

## CDCP1 promotes activation of STAT3 signaling

To dissect the signaling pathways downstream of the CDCP1-Src axis, we performed DNA microarray analyses of MDCK cysts expressing either wild-type CDCP1 or the CDCP1-CG mutant. Gene ontology analysis indicated that CDCP1 expression is involved in the regulation of signal transduction (Fig S5A) and Ingenuity Pathway Analysis revealed that the STAT3 pathway was more prominently activated by CDCP1 than by CDCP1-CG expression (Fig S5B and C and Tables S2 and S3). The level of phosphorylated STAT3 (pY705) in-creased markedly in cysts expressing wild-type CDCP1 (Fig 3A). Furthermore, quantitative PCR analysis confirmed that the ex-pression of several matrix metalloproteinase (MMP)-encoding genes was increased after up-regulation of CDCP1 (Fig 3B). Im-munofluorescent staining of laminin revealed that CDCP1 expres-sion disrupted the basement membrane, allowing the extension of cell protrusions (Fig S6A). This effect was suppressed by treatment with marimastat, a pan-MMP inhibitor (Fig S6B and C). Similar marimastat-mediated suppression of cell protrusion extension was also observed in HGF-stimulated MDCK cysts (Fig S6D–F). Analysis with DQ-collagen, a fluorescent indicator of collagen degradation, also showed that CDCP1 expression induced degradation of the ECM (Fig S6G). These observations suggest that the HGF-induced for-mation of cell protrusions is associated with ECM rearrangement via STAT3-induced up-regulation of MMPs. The expression of mitogenic genes such as *MYC* (Myc) and *CCND1* (Cyclin D1) was also up-regulated in cysts expressing CDCP1 (Fig 3B), indicating that CDCP1-induced cell proliferation is attributable to STAT3 activation.

We confirmed the contribution of STAT3 to cell protrusion for-mation and proliferation by specifically perturbing its activity. Activation of STAT3 using the STAT3-MER system induced the for-mation of cell protrusions and a multi-layered structure (Fig S6H–J). In contrast, treatment of MDCK cysts with STAT3-specific inhibitors suppressed CDCP1-induced cell growth (Fig 3C and Table S1), and overexpression of dominant negative STAT3 (STAT3-Y705F) also

inhibited CDCP1-induced cellular events (Fig 3D and E). Furthermore, STAT3 inhibitors efficiently suppressed cell proliferation (Fig 3F and G) and protrusion formation (Fig 3H) in HGF-stimulated MDCK cysts, underscoring the crucial role of STAT3 activation in HGF signaling.

## CDCP1 focally integrates Src and Met-STAT3 signaling

We also found that the morphological changes induced by over-expression of Met were suppressed after *CDCP1*-knockout in MDCK cells (Fig S7A), suggesting the crucial role of CDCP1 in Met functions. Accordingly, we investigated the functional link between Met and CDCP1. An inhibitor screening assay showed that various Met-specific inhibitors potently suppressed CDCP1-induced pheno-typic changes (Fig 4A and Table S1), indicating that Met activity is involved in the function of CDCP1. To assess the physical interaction between these proteins, Met and various CDCP1 mutants that lacked specific extracellular CUB domains were co-expressed in HEK293 cells (Fig 4B and C). Co-immunoprecipitation assays revealed that CDCP1 interacted with Met independently of HGF stimulation. This HGF-independent CDCP1-Met interaction was also observed in the renal cancer cell lines, A498 and ACHN, in which endogenous CDCP1 and Met are up-regulated (Fig S7B). Furthermore, the CDCP1-Met interaction was enhanced by removal of the first CUB domain of CDCP1 but diminished after deletion of all the CUB domains (Fig 4B and C), indicating that CDCP1 interacts with Met through extra-cellular domains, and that removal of the first CDCP1 CUB domain is necessary for efficient association with Met.

Because CDCP1 is cleaved between the first and second CUB domains (32, 33), it is likely that cleaved CDCP1 functionally interacts with Met. To verify this hypothesis, we analyzed the function of a mutant CDCP1 (CDCP1-PR) that was resistant to proteolytic shedding owing to the presence of three point mutations within the protease recognition sites (K365A-R368A-K369A) (Fig 4D). Expression of CDCP1-PR induced the activation of Src to a level similar to that observed with wild-type CDCP1 but failed to activate STAT3 (Fig 4D). Consistent with these biochemical effects, CDCP1-PR failed to in-duce the formation of cell protrusions (Fig 4E and F). These data suggest that an association between cleaved CDCP1 and Met is required for Src-mediated STAT3 activation and subsequent cel-lular events.

To elucidate the molecular mechanisms of CDCP1-mediated activation of Met-STAT3 signaling, we analyzed the behavior of Met proteins in CDCP1-overexpressing cells. After CDCP1 over-expression, total Met protein levels gradually increased (Figs 5A and S8A), translocated from the intracellular compartment to the

---

(50 ng/ml) for the indicated time periods. Activated Src (pY418) was visualized with an Alexa Fluor 488–conjugated antibody (green), and actin filaments were stained with Alexa Fluor 594-phalloidin (magenta); arrowheads indicate transiently formed protrusions. **(F)** mCherry-GPI–overexpressing MDCK cysts embedded within the collagen matrix were incubated in the presence of HGF (50 ng/ml) for the indicated time periods. Actin filaments were stained with Alexa Fluor 488-phalloidin (green); arrowheads indicate transiently formed protrusions. **(G)** MDCK cysts embedded within the collagen matrix were incubated in the presence of HGF (50 ng/ml) for the indicated time periods. Localization of CDCP1 was visualized using an Alexa Fluor 488–conjugated antibody (green), and actin filaments were stained with Alexa Fluor 594-phalloidin (magenta); arrowheads indicate transiently formed protrusions. **(H, I, J)** Wild-type and *CDCP1*-knockout MDCK cysts were incubated in the presence of HGF (50 ng/ml) for 1 d. **(H)** Ki67 was visualized with an Alexa Fluor 488-conjugated antibody (green), and actin filaments were stained with Alexa Fluor 594-phalloidin (magenta); arrowheads indicate transiently formed protrusions. **(I)** Diameter ($\mu$m) of cysts (n = 100). **(J)** Fraction of the total number of cysts counted (n > 100) with protrusions. Scale bars: 10 $\mu$m. Data information: In (B, C, D, I, J), the mean ratios ± SD were obtained from three independent experiments. *$P$ < 0.05; **$P$ < 0.01; ***$P$ < 0.001; ****$P$ < 0.0001; NS, not significantly different; two-way ANOVA was calculated compared with HGF-treated cysts.

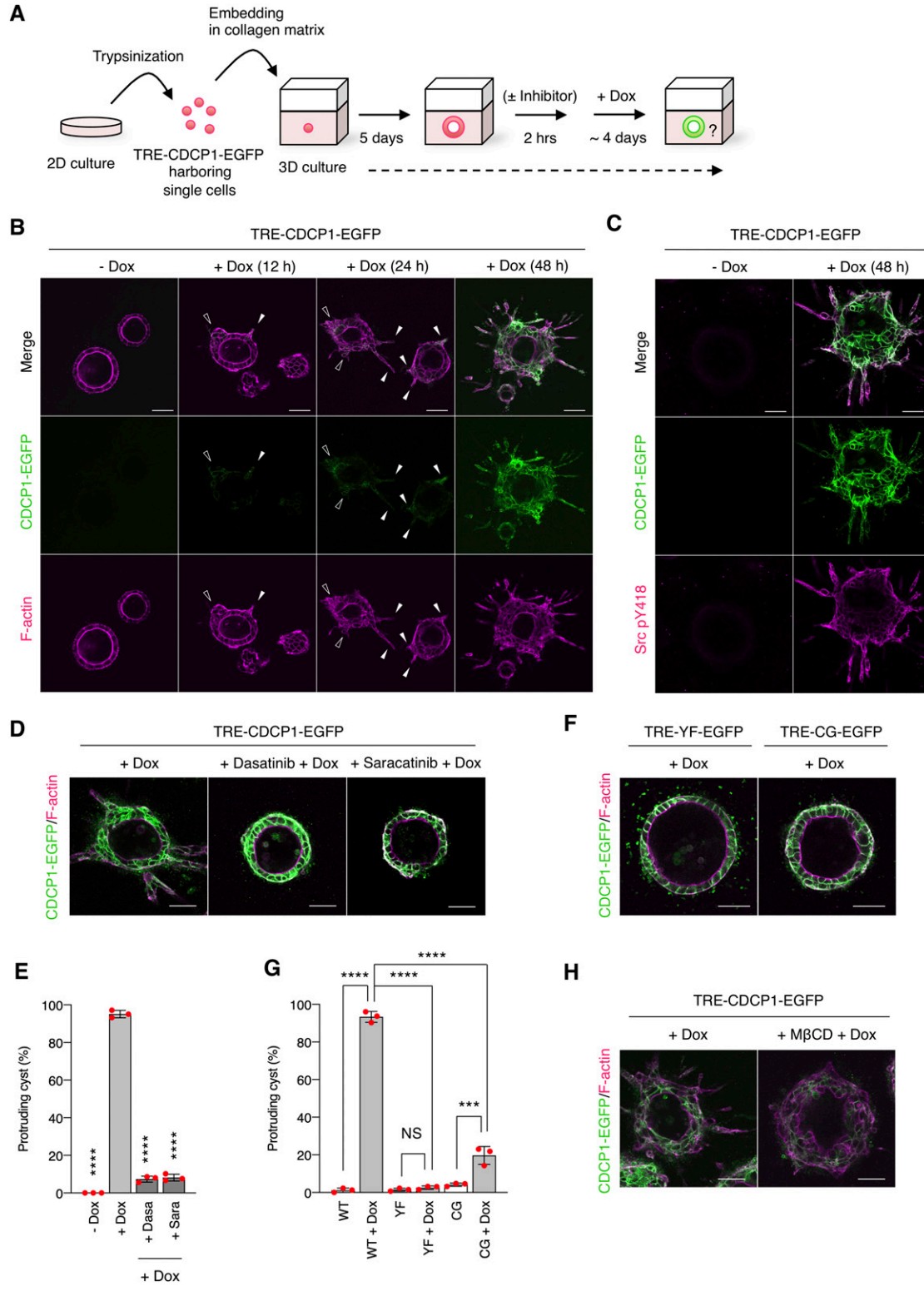

**Figure 2.  Up-regulation of CDCP1 induces hepatocyte growth factor-related phenotypic changes by activating Src in lipid rafts.**
**(A)** Schematic illustration of analysis of TRE-CDCP1-EGFP-harboring MDCK cysts (CDCP1-EGFP cysts). TRE-CDCP1-EGFP-harboring cells were cultured within a collagen matrix for 5 d for cyst formation, and then CDCP1-EGFP cysts were incubated in the presence of Dox for ~4 d. **(B)** CDCP1-EGFP cysts embedded within the collagen matrix were incubated in the presence of Dox (1 μg/ml) for the indicated time periods. Arrowheads indicate protruding cells and open arrowheads indicate multi-layered structures. **(C)** CDCP1-EGFP cysts were incubated in the presence of Dox (1 μg/ml) for 2 d. Activated Src was visualized with a Src pY418 antibody (magenta). **(D, E)** CDCP1-EGFP cysts embedded within the collagen matrix were pretreated with 20 nM dasatinib or 10 μM saracatinib for 2 h, and then incubated with Dox (1 μg/ml) for 4 d.

plasma membrane (Fig 5B and C), and partly relocated into the DRM fractions (Fig S8D). Furthermore, up-regulated Met was phosphorylated at Y1234/1235 and Y1349, sites that are necessary for its kinase activity and STAT3 association, respectively (Figs 5A and S8B and C). In contrast, Met accumulation was not induced by overexpression of CDCP1-YF or CDCP1-PR mutants (Figs 5A and S8A) and was inhibited by treatment with saracatinib (Figs 5B–D and S8E–G). These findings suggest that activated Met is trapped by the cleaved CDCP1-Src complex and accumulates on the plasma membrane. Furthermore, upon CDCP1 overexpression, phosphorylated Met interacted with STAT3, but not phosphorylated STAT3, and this association was disrupted by treatment with saracatinib and Met/Ron kinase inhibitor (Fig 5E and F). This indicates that the CDCP1-Src complex is required for efficient association between phosphorylated Met and STAT3. Taken together, these biochemical data suggest that CDCP1-Src in lipid rafts promotes Met-mediated STAT3 activation (Figs 5G and S8H).

### Compensatory renal growth is attenuated in Cdcp1-knockout mice

To investigate the role of CDCP1-mediated spatial regulation of Met-STAT3 signaling in compensatory renal growth following UNX, we generated $Cdcp1$-knockout ($Cdcp1^{-/-}$) mice in a C57/BL6 background using the CRISPR/Cas9 system (Fig S9A–D). $Cdcp1^{-/-}$ mice grew normally and did not show any overt phenotype under normal conditions (Fig S9E and F), as reported previously (34). 8 wk after UNX, the remaining kidney was enlarged in wild-type and heterozygous $Cdcp1^{+/-}$ mice (Fig 6A). Kidney/body weight ratios in wild-type and $Cdcp1^{+/-}$ mice were elevated to ~136% and 135% relative to those in sham-operated mice, respectively. In contrast, the kidney/body weight ratio elevation was significantly decreased in $Cdcp1^{-/-}$ mice (~121%) (Figs 6B and S10A–C). However, no prominent histopathological difference was observed among the kidneys of all mice by microscopic observation (Figs 6C and S10D). As compensatory renal growth is achieved via the expansion of proximal renal tubules (3), we visualized the renal proximal tubules using fluorescein-labeled specific lectin (LTL-FITC) and the surrounding basement membrane that was mainly comprised of collagen IV (Fig 6D). Wild-type mice exhibited prominent thickening of LTL-positive proximal renal tubules (~148%), whereas this thickening was suppressed in $Cdcp1^{-/-}$ mice (~117%) (Fig 6D and E). Furthermore, enlargement of the proximal tubule area was suppressed by loss of Cdcp1 (wild-type: 121% versus $Cdcp1^{-/-}$: 112%) (Fig 6D and F). However, enlargement of the glomerulus area was not significantly different between wild-type and $Cdcp1^{-/-}$ mice (116% versus 114%, respectively) (Fig 6C and G). These morphometric data were consistent with the observed elevation of the kidney/body weight ratios (Fig 6B), confirming that compensatory growth of proximal renal tubules is suppressed in $Cdcp1^{-/-}$ mice.

To determine the cause of defective renal growth in $Cdcp1^{-/-}$ mice, we analyzed the remaining kidney at earlier stages of compensatory growth (within 4 d after UNX) because the expression of HGF and Met is transiently up-regulated in renal tissues within 12 h (8, 9). The mass of the remaining kidney immediately increased in wild-type mice, whereas acute enlargement of the kidney tended to be delayed in $Cdcp1^{-/-}$ mice (Fig 7A and B), indicating that cell growth/hypertrophy of proximal tubules after nephrectomy (within 2 d) was attenuated by loss of Cdcp1. Immunofluorescence analysis revealed that activation of Met (pY1234/1235) and STAT3 (pY705) occurred in wild-type renal tubules 12 h after UNX, as reported previously (14), whereas the transient activation of both signals was appreciably attenuated in $Cdcp1^{-/-}$ renal tubules (Fig 7C and D). Activation of CDCP1 (pY734) also occurred in renal tubules in a manner similar to that of Met and STAT3 (Fig 7E). Notably, activated Met, STAT3, and CDCP1-Src co-localized in a subset of intracellular small vesicles that were positive for early endosome antigen 1 (EEA1) (Fig S11A–C), a marker of early endosomes, supporting the intimate interactions among these signaling molecules. These results suggest that the CDCP1-Src axis mediates Met-STAT3 signaling even in the proximal renal tubules.

We also analyzed cell proliferation/hyperplasia by immunostaining for Ki67. The ratio of Ki67-positive cells in wild-type proximal tubules gradually increased and peaked 2 d after UNX (Fig 7F and G). However, in $Cdcp1^{-/-}$ mice, the appearance of Ki67-positive cells was delayed, and the overall number of proliferating cells was decreased. These results suggest that the HGF-induced cell proliferation through Met-STAT3 signaling is also attenuated in $Cdcp1^{-/-}$ mice, thereby retarding the onset of compensatory renal growth (Fig 7H).

Finally, we scrutinized the surrounding environment of the proximal tubules during the compensatory growth. Before UNX, collagen IV was observed around the proximal tubules in both wild-type and $Cdcp1^{-/-}$ mice (Figs 6D and S11D and E). In wild-type mice, surrounding collagen IV was transiently degraded 2 d after UNX, whereas this degradation was attenuated in $Cdcp1^{-/-}$ mice. To identify the molecules involved in transient degradation of the basement membrane, we focused our analysis on secretory MMPs, including MMP2 and MMP9, both of which are targets of STAT3 and are expressed in the proximal tubules (35). In wild-type mice, MMP2 and MMP9 were up-regulated and occasionally concentrated in vesicle-like structures in proximal tubules 24 h after UNX (Fig S11F and G). However, these phenomena were barely detectable in the $Cdcp1^{-/-}$ proximal tubules. These observations suggest that MMPs contribute to compensatory renal growth downstream of CDCP1-Src–mediated Met-STAT3 signaling (Fig 7H).

## Discussion

To determine the regulatory mechanisms underlying HGF-induced regenerative renal growth, we dissected HGF signaling using MDCK cysts as in vitro models of renal tubules, and identified Src and its scaffold protein, CDCP1, as regulatory elements of HGF signaling. Mechanistic analysis showed that CDCP1-Src is required for the

**(E)** Fraction of the total number of cysts counted (n > 150) with protrusions. **(F, G)** CDCP1-YF-EGFP and CDCP1-CG-EGFP cysts embedded within the collagen matrix were incubated in the presence of Dox (1 μg/ml) for 4 d. **(G)** Fraction of the total number of cysts counted (n > 150) with protrusions. **(H)** CDCP1-EGFP cysts embedded within the collagen matrix were pretreated with 1 mM MβCD for 2 h, and then incubated with Dox (1 μg/ml) for 4 d. Actin filaments were stained with Alexa Fluor 594-phalloidin (magenta). Scale bars: 50 μm. Data information: In (E, G), the mean ratios ± SD were obtained from three independent experiments. ***P < 0.001; ****P < 0.0001; NS, not significantly different; two-way ANOVA compared with the Dox-treated CDCP1-EGFP cysts.

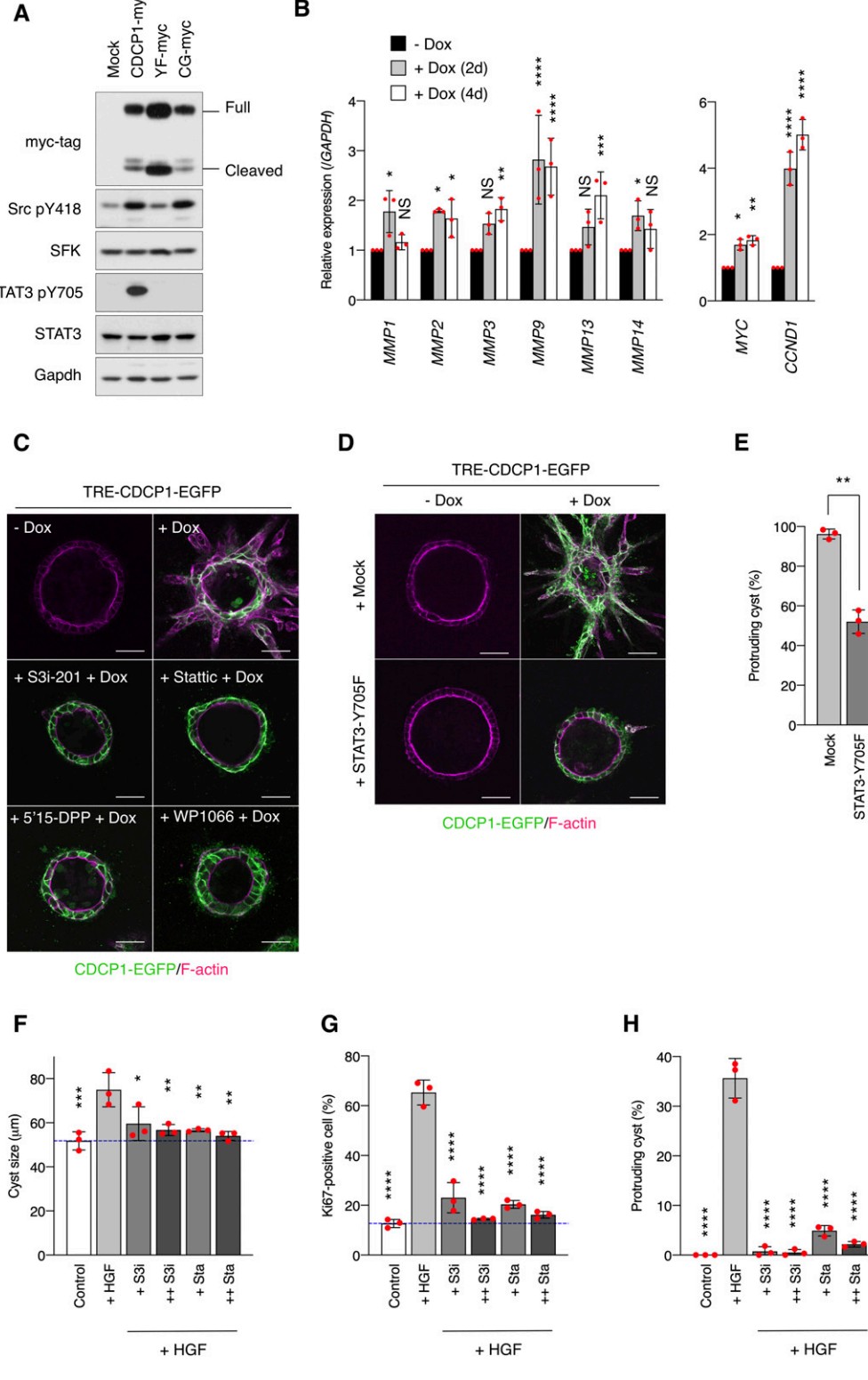

**Figure 3. STAT3 activation is required for CDCP1-induced phenotypic changes.**
**(A)** CDCP1-myc– and mutant-overexpressing MDCK cells were embedded within the collagen matrix and cultured for 9 d. Cyst lysates were subjected to immunoblotting using the indicated antibodies. **(B)** CDCP1-EGFP cysts were incubated with Dox (1 μg/ml) for 2 or 4 d, and then subjected to quantitative real-time PCR. Relative mRNA expression levels were calculated by setting the mean value for non-treated cysts to one. **(C)** CDCP1-EGFP cysts were pretreated with the indicated STAT3-specific inhibitors for 2 h and then incubated with Dox (1 μg/ml) for 4 d. **(D, E)** STAT3-Y705F–overexpressing CDCP1-EGFP cysts were incubated with Dox (1 μg/ml) for 4 d. **(D)** Actin filaments were stained with Alexa Fluor 594-phalloidin (magenta). Scale bars: 50 μm. **(E)** Fraction of the total number of cysts counted (n > 150) with protrusions. **(F, G, H)** MDCK cysts embedded within the collagen matrix were pretreated with S3i-201 (+ S3i, 50 nM; ++ S3i, 100 nM) or stattic (+ Sta, 2.5 nM; ++ Sta, 5.0 nM) for 2 h and incubated in the presence of hepatocyte growth factor (50 ng/ml) for 1 d. **(F)** Diameter (μm) of cysts (n = 100). **(G)** Ratio of Ki67-positive cell in cyst (n = 50). **(H)** Fraction of the total number of cysts counted (n > 100) with protrusions. Dotted blue indicates the average diameter of non-treated cysts. Data information: In (B, E, F, G, H), the mean ratios ± SD were obtained from three independent experiments. *P < 0.05; **P < 0.01; ***P < 0.001; ****P < 0.0001; NS, not significantly different; ANOVA was calculated compared with the non-treated cyst (B) or the hepatocyte growth factor-treated cysts (F, G, H); unpaired two-tailed t test (E). Source data are available for this figure.

spatiotemporal activation of the HGF-Met-STAT3 pathway in lipid rafts, leading to the promotion of cell growth/proliferation and morphological changes. To verify the role of CDCP1 in vivo, we examined the effects of Cdcp1 ablation on compensatory renal growth after UNX. Immunohistological analysis revealed that Cdcp1 ablation suppressed transient activation of Met, Src, STAT3, and MMPs in proximal renal tubules, resulting in suppression of compensatory renal growth. Although the experiments involved transient

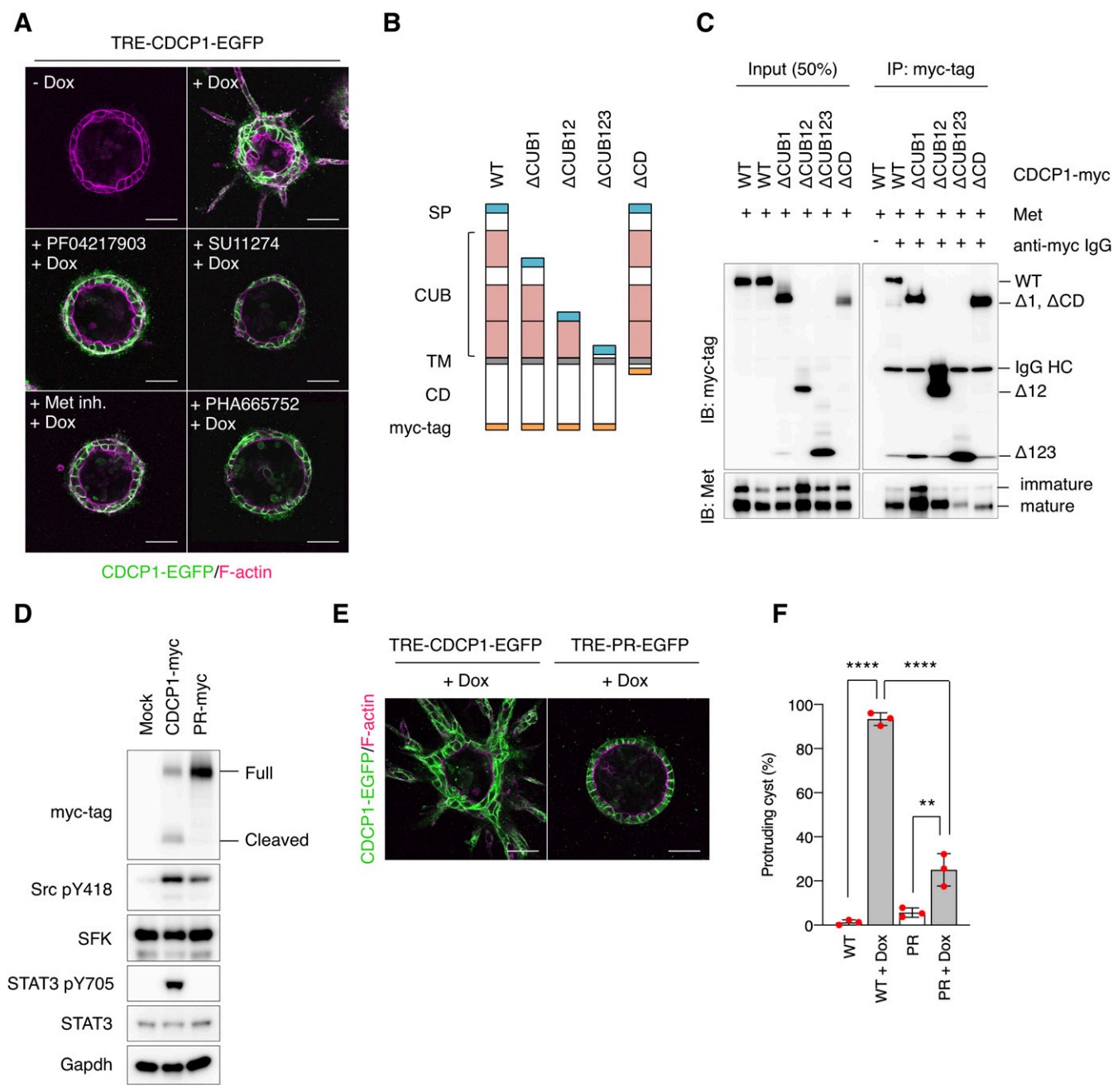

**Figure 4. CDCP1-Met association is required for CDCP1-induced phenotypic changes.**
**(A)** CDCP1-EGFP cysts were pretreated with the indicated Met-specific inhibitors for 2 h and then incubated with Dox (1 μg/ml) for 4 d. **(B, C)** Schematic representation of CDCP1 and the deletion mutants. **(B)** SP, signal peptide; CUB, CUB domain; TM, transmembrane domain; CD, cytosolic domain. Lysates from HEK293 cells overexpressing both CDCP1-myc and Met were subjected to immunoprecipitation with an anti-myc-tag antibody. Immunoprecipitates were subjected to immunoblotting using the indicated antibodies. IgG HC, IgG heavy chain. **(D)** CDCP1-myc- and CDCP1-PR-myc-overexpressing MDCK cells were embedded within the collagen matrix and cultured for 9 d. Cyst lysates were subjected to immunoblotting analysis using the indicated antibodies. **(E, F)** CDCP1-PR-EGFP cysts embedded within the collagen matrix were incubated with Dox (1 μg/ml) for 4 d. **(E)** Actin filaments were stained with Alexa Fluor 594-phalloidin (magenta). Scale bars: 50 μm. **(F)** Fraction of the total number of cysts counted (n > 150) with protrusions. Data information: In (F), the mean ratios ± SD were obtained from three independent experiments. **$P < 0.01$; ****$P < 0.0001$; NS, not significantly different; two-way ANOVA was calculated compared with the Dox-treated cysts. Source data are available for this figure.

activation and/or lower expression levels of endogenous Cdcp1 in the kidney, these in vivo observations together with the in vitro data underscore the crucial role of CDCP1-Src in the regulation of the HGF-Met-STAT3 pathway.

Upon stimulation with HGF, CDCP1 was temporally concentrated at the tip of the protruding cells in MDCK cysts. As previous reports have shown that CDCP1 expression is induced by HGF (36) or by EGF-mediated activation of the MAPK pathway (37, 38), it is likely that the

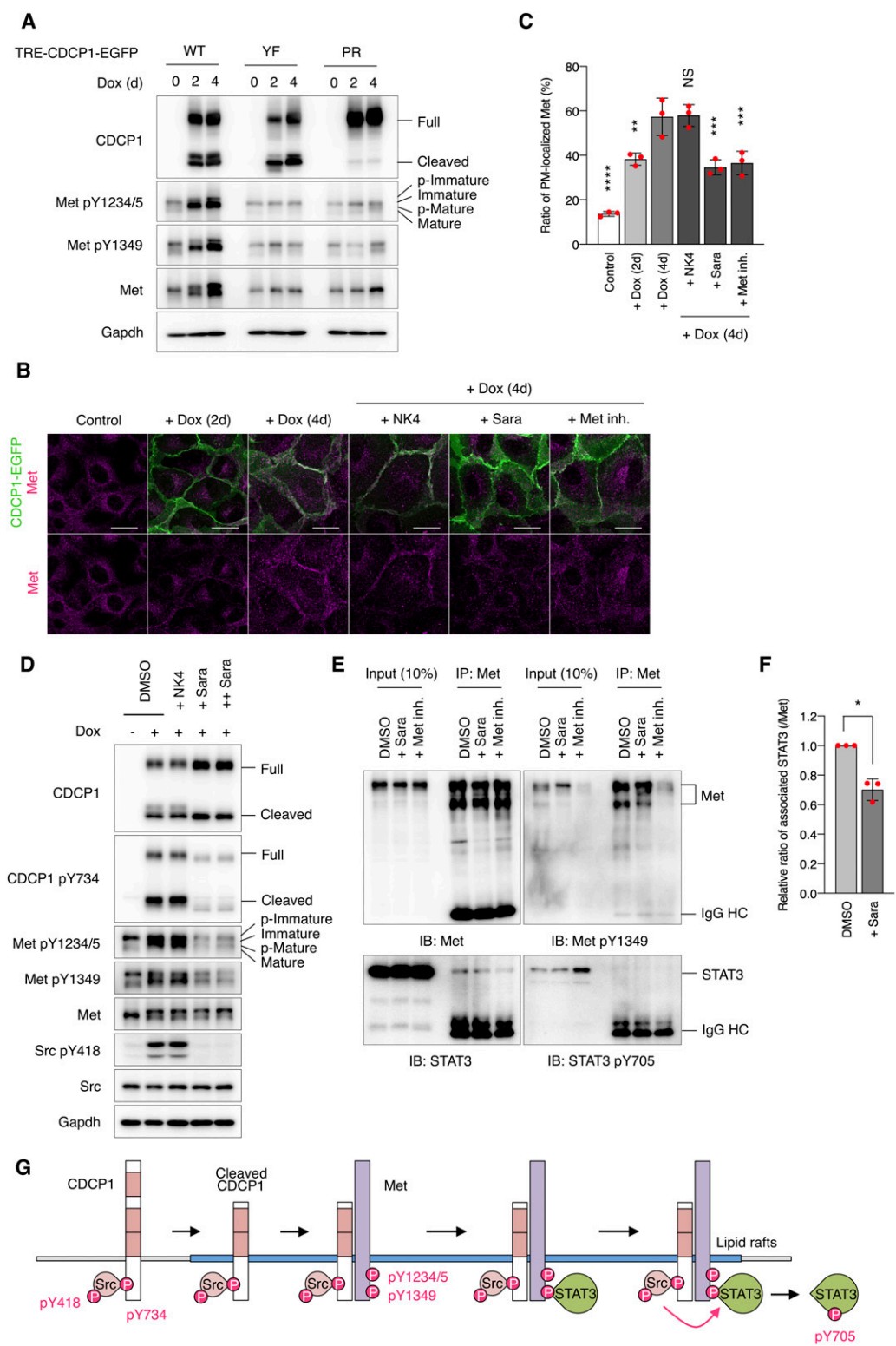

**Figure 5. CDCP1 regulates Met–STAT3 activation in lipid rafts.**
**(A)** CDCP1-EGFP, CDCP1-YF-EGFP, and CDCP1-PR-EGFP cells were incubated with Dox (1 µg/ml) for 2, 4 d. Cell lysates were subjected to immunoblotting analysis using the indicated analysis using the indicated antibodies. **(B, C)** CDCP1-EGFP cells were incubated with Dox (1 µg/ml) for 2, 4 d. NK4 (1 µg/ml) were added 2 h before the Dox treatment. Saracatinib (10 µM, Sara) or Met/Ron kinase inhibitor (100 nM, Met inh.) were added 3 d after the Dox treatment. Met were stained with Alexa Fluor 594-phalloidin (magenta). Scale bars: 20 µm. **(C)** Ratio of plasma membrane-localized Met was calculated (n = 50). **(D)** CDCP1-EGFP cells were incubated with Dox (1 µg/ml) for 4 d. NK4 (1 µg/ml) were added 2 h before the Dox treatment. Saracatinib (+ Sara, 10 µM; ++ Sara 20 µM) were added 3 d after the Dox treatment. Cell lysates were subjected

observed up-regulation of CDCP1 was caused by activation of the HGF-Met-MAPK pathway. We also found that continuous up-regulation of CDCP1 induced the formation of multiple cell protrusions and cell growth/proliferation through Met activation without HGF stimulation. Notably, when CDCP1 expression was terminated by removing Dox from the media, cells in the protruding cord-like structures recovered their epithelial features, resulting in the formation of a luminal structure (Fig S12A). This finding suggests that the temporal up-regulation of CDCP1 by growth factors such as HGF may trigger the initial phase of morphogenesis, specifically cell protrusion extension and growth, and that a coordinated negative feedback regulation of CDCP1 may be involved in the establishment of the compound tubular/acinar epithelial system.

In vitro analysis revealed that HGF-mediated cell growth, proliferation, and morphological changes required activation of mTOR and Src signaling. Inhibition of mTOR signaling suppressed cell growth/proliferation, but not morphological changes. Given that mTORC2 regulates actin cytoskeleton organization through activation of the PKC pathway (39), mTOR inhibitors may interfere with actin organization and disrupt cytoskeleton dynamics. Nonetheless, cell protrusion formation was not inhibited. In contrast, MMP inhibitors suppressed morphological changes, but not cell growth and proliferation. These observations suggest that HGF-mediated phenotypic changes are mediated by both mTOR signaling and the STAT3-MMP axis. As such, CDCP1-mediated morphological changes were efficiently suppressed by concurrent treatment with mTOR and MMP inhibitors (Fig S12B).

In our model system, CDCP1 was tyrosine-phosphorylated by Src and in turn trapped the activated Src mainly in the lipid rafts. Although CDCP1-mediated Src activation has been reported in other cell types (40, 41), we demonstrated here that localization of the CDCP1-Src complex in lipid rafts is required for HGF-induced phenotypic changes in MDCK cysts. Furthermore, although STAT3 phosphorylation by activated Src has been reported previously (42, 43), we found that CDCP1 mediates Src-induced STAT3 phosphorylation in lipid rafts. We also found that the Src-induced STAT3 activation requires a physical interaction between cleaved CDCP1 and Met. Because STAT3 interacts directly with activated Met through an SH2 domain (13), it is likely that the cleaved form of CDCP1 binds Src and interacts with the Met-STAT3 complex in lipid rafts to allow efficient phosphorylation of STAT3 by Src (Fig 5G). This focal integration of the CDCP1-Src axis with the Met-STAT3 complex in lipid rafts may be crucial for the activation of STAT3, which is required for HGF function. CDCP1 also activates PKCδ through interaction with its Tyr762 residue (44). In our experimental system, overexpression of the CDCP1-Y762F mutant induced protrusion formation (Fig S12C), and CDCP1-induced protrusion formation was not suppressed by a PKC inhibitor (Fig S12D). These results suggest that PKCδ is not involved in CDCP1-mediated morphological

changes. CDCP1 also interacts with other transmembrane proteins, including EGFR, HER2, integrin β1, and E-cadherin (45, 46, 47, 48). However, CDCP1- or HGF-induced morphological changes were not suppressed by potent EGFR and HER2 inhibitors (Fig S13 and Table S1), suggesting that both these receptor proteins may not be involved in the signaling required for the morphological changes. These findings indicate that CDCP1 may function as a more general regulator of membrane receptor signaling, and that its contribution may depend on various conditions, including expression levels, modification of partner proteins, cell characteristics, and surrounding environment.

We discovered that overexpression of CDCP1 up-regulated Met phosphorylation and translocated it to the plasma membrane. This process was not inhibited by pretreatment with NK4 (see also Fig S12E and F), indicating that CDCP1-mediated Met phosphorylation is independent of HGF, raising the possibility that up-regulation of Met phosphorylation may be induced via a novel mechanism. Two models could potentially explain how CDCP1 up-regulates Met phosphorylation even in the absence of HGF. First, CDCP1-bound Src directly phosphorylates Met Tyr1234/1235 and Tyr1349 residues, as reported previously (49, 50). In support of this, saracatinib inhibited not only Src activation but also CDCP1-mediated Met phosphorylation, implying that CDCP1 supports Src-mediated Met phosphorylation. Second, CDCP1 traps spontaneously phosphorylated Met. Met phosphorylation is known to be induced not only by HGF but also by several other stimuli (51, 52). We also observed that serum depletion from media caused a reduction in Met phosphorylation (Fig S8I–K), suggesting that Met may be phosphorylated by other stimulatory factors. Furthermore, CDCP1-dependent Met phosphorylation required a longer period (in days) than that by HGF stimulation (in minutes), during which Met was translocated from the intracellular compartment to the plasma membrane with the ratios of p-Met/Met being kept constant. These observations suggest that phosphorylated Met is trapped and/or stabilized on the plasma membrane via CDCP1. It is also likely that these two mechanisms may occur synchronously. Further investigation is needed to understand the detailed molecular mechanisms of CDCP1-induced up-regulation of Met phosphorylation.

Up-regulation of CDCP1 has been implicated in tumor progression (32, 38, 45, 53, 54, 55, 56, 57, 58, 59, 60, 61, 62). In some cancer cells, up-regulated CDCP1 promotes invasion, metastasis, and tumor growth (33, 40, 45, 58, 60, 63, 64, 65), although the underlying mechanisms remain elusive. Our in vitro findings presented here suggest that CDCP1-mediated activation of the Src-STAT3 pathway contributes to malignant progression by inducing invasion-like and growth-promoting phenotypes, even in cancer cells. Indeed, we found that overexpression of CDCP1 in MDCK cysts induced the expression of cytokeratin 14 (Fig S12G), a promising marker of collective invasion of cancer cells (66), suggesting a potential role

---

to immunoblotting analysis using the indicated analysis using the indicated antibodies. **(E, F)** CDCP1-EGFP cells were incubated with Dox (1 μg/ml) for 4 d. Saracatinib (10 μM, Sara) were added 3 d after the Dox treatment. Met/Ron kinase inhibitor (100 nM, Met inh.) were added 30 min before cell extraction. Cell lysates were subjected to immunoprecipitation with an anti-Met antibody, and immunoprecipitates were subjected to immunoblotting using the indicated antibodies. IgG HC, IgG heavy chain. **(F)** Relative ratio of associated STAT3 was calculated by setting the mean value for non-treated cells to one. **(G)** Schematic diagram of the CDCP1-Src-Met complex-mediated STAT3 activation. Data information: In (C, F), the mean ratios ± SD were obtained from three independent experiments. *P < 0.05; **P < 0.01; ***P < 0.001; ****P < 0.0001; NS, not significantly different; two-way ANOVA was calculated compared with the Dox-treated cells (C); unpaired two-tailed t test (F). Source data are available for this figure.

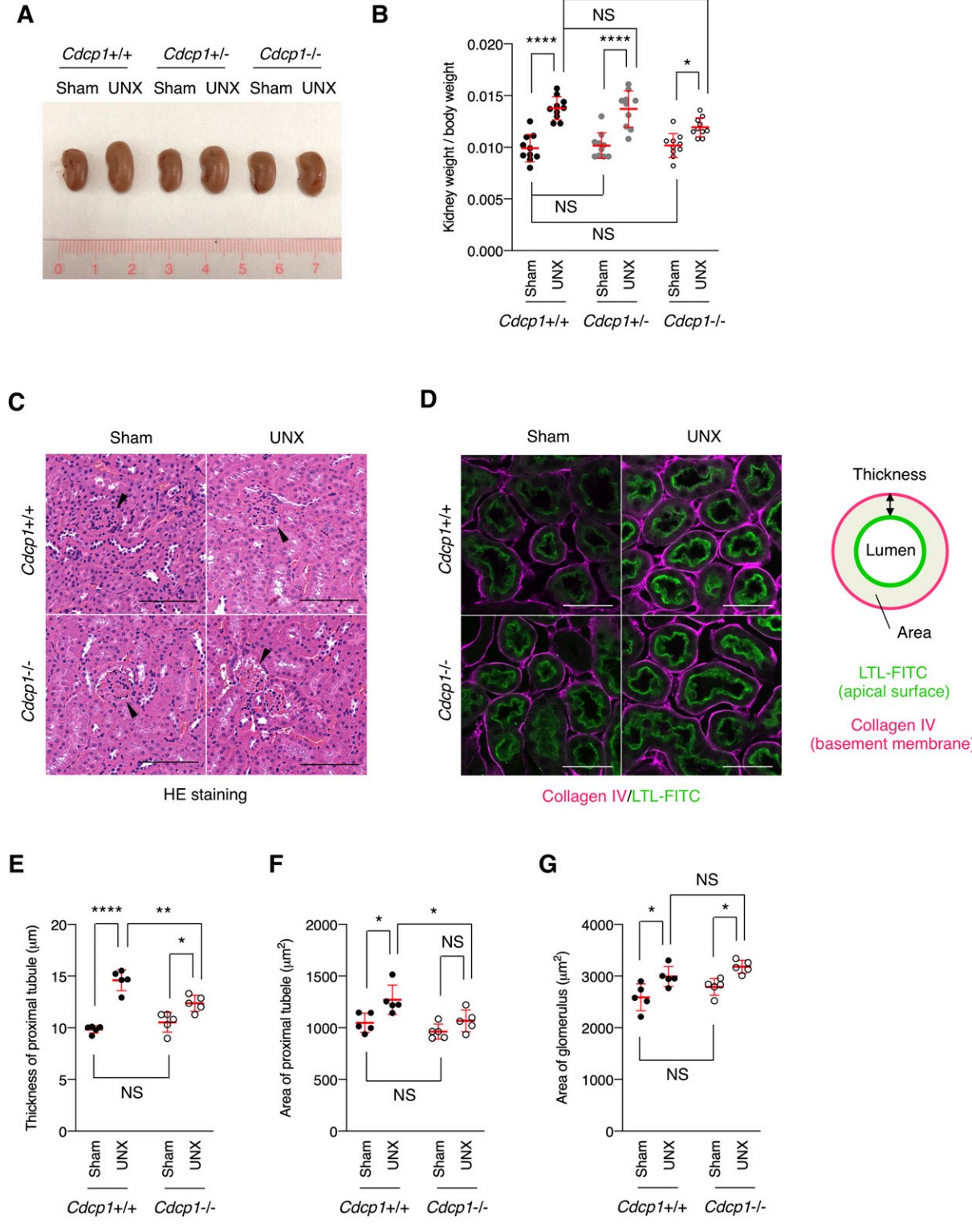

**Figure 6. Compensatory renal growth after UNX is suppressed in *Cdcp1*-knockout mice.**
**(A, B)** *Cdcp1* wild-type (+/+), heterozygous (+/−), and homozygous knockout (−/−) mice at 8 wk of age were subjected to UNX or sham operation. Regenerative growth of remaining kidney was analyzed 8 wk after operation and increases in remaining kidney/body weight ratios were assessed. **(C)** The remaining kidney was removed from UNX- or sham-operated mice, and was subjected to hematoxylin-eosin (HE) staining; arrowheads indicate glomeruli. **(D, E, F)** The remaining kidney after UNX was subjected to microscopic immunofluorescence analysis with specific antibodies against collagen IV (magenta), and proximal tubules were visualized with FITC-LTL (green). Scale bars: 50 $\mu$m. Thickness (E) and area (F) of proximal tubules were determined by the distance ($\mu$m, n = 100) and area ($\mu$m$^2$, n = 50) between FITC-LTL-stained lumen and collagen IV-enriched basement membrane as depicted in panel (D). **(G)** Area ($\mu$m$^2$) of glomerulus in the remaining kidney was evaluated by using HE-staining images in panel (C) (n = 50). Data information: In (B, E, F, G), the mean ratios ± SD were obtained from 10 (B) or five mice (E, F, G) per group. *$P$ < 0.05; **$P$ < 0.01; ****$P$ < 0.0001; NS, not significantly different; two-way ANOVA.

**Figure 7. Met-STAT3 signaling is attenuated in *Cdcp1*-knockout mice.**
**(A, B)** *Cdcp1* wild-type (+/+) and homozygous knockout (−/−) mice at 8 wk of age were subjected to UNX or sham operation. Compensatory growth of remaining kidney was analyzed at the indicated time points after operation and increases in remaining kidney/body weight ratios were assessed. **(C, D, E, F)** The remaining kidney after UNX was subjected to microscopic immunofluorescence analysis with specific antibodies against Met pY1234/1235 (C), STAT3 pY705 (D), CDCP1 pY734 (E), and Ki67 (F) and Alexa Fluor 594–conjugated secondary antibody (magenta). Proximal tubules were visualized by staining with FITC-LTL (green). Representative images were shown. Scale bars: 50 µm. **(G)** Ki67-positive proximal tubules (%) were estimated by calculating the ratio of Ki67-stained tubules to the total number of tubules (n > 100). **(H)** Schematic model

of up-regulation of the CDCP1-Src axis in this process. Further analyses of the functions of the CDCP1-Src-Met-STAT3 pathway in a wide range of cancer cells may reveal potentially new therapeutic targets for the treatment of some malignant cancers.

HGF is up-regulated immediately after the loss of kidney mass and plays important roles in compensatory renal growth through the induction of cell growth, proliferation, and anti-apoptotic effects (12). In this study, we uncovered the contribution of CDCP1-Src in HGF-induced compensatory renal growth in Cdcp1-deficient mice. Immunofluorescence analyses showed that transient activation of Met and STAT3 in renal tubule cells after UNX was clearly attenuated in Cdcp1-deficient mice. Furthermore, up-regulation of MMP2/9 and transient alteration of basement membrane collagen IV were diminished in the renal tubule cells of Cdcp1-deficient mice after UNX (Fig S11H). These results are mechanistically consistent with those of our in vitro studies using MDCK cysts, supporting the crucial role of the functional integration of CDCP1-Src with Met-STAT3 pathways, even during compensatory renal growth. Although up-regulation of MMPs and related factors after UNX has been reported in previous studies, their contribution to compensatory renal growth has remained unknown (67, 68, 69). Similar observations have been reported in patients with compensatory renal growth (70). Transient alteration of basement membrane integrity during the early stage of compensatory growth may be required for efficient cell growth/hypertrophy and proliferation/hyperplasia of renal tubules (e.g., arrangement of space expansion). These findings may shed new light on understanding of the molecular details of compensatory organ growth. However, renal growth was not completely suppressed by CDCP1 loss, suggesting that other CDCP1-related molecules or distinct growth factor signaling pathways may compensate for the function of CDCP1 in vivo. Given that the expression of CDCP1 is temporally upregulated under pathological conditions such as tissue injury, hypoxia, and cancer recurrence (32), it is likely that upregulated CDCP1 may function to acutely amplify the HGF signaling that is required for regenerative organ growth in critical situations. Furthermore, because STAT3 is an important regulator of the regeneration of other organs such as the liver, intestine, and muscle (71, 72, 73), it is also possible that the temporal up-regulation of CDCP1 contributes more strongly to the promotion of organ regeneration by inducing focal activation of the Src-STAT3 pathway.

In conclusion, we found that CDCP1 serves as a regulator of HGF-induced compensatory renal growth by focally integrating Src and Met-STAT3 pathways. Our discovery of the CDCP1-Src axis as a new component of HGF-Met signaling provides insights into the regulatory mechanisms underlying the multifaceted functions of HGF during morphogenesis and/or regenerative growth, and might contribute to the development of more promising therapies for

malignant cancers that are associated with the up-regulation of CDCP1 and Met.

# Materials and Methods

## Cell culture

MDCK type I, HEK293T, A498, and ACHN cells were cultured in DMEM supplemented with 10% FBS at 37°C in a 5% $CO_2$ atmosphere. Three-dimensional culture was performed using collagen type I (Cell-matrix Type I-A; Nitta Gelatin) according to the manufacturer's protocol. Collagen type I (3 mg/ml) was neutralized with reconstitution buffer (2.2% $NaHCO_3$, 0.05 N NaOH and 200 mM Hepes) and diluted with 5× DMEM (Gibco). MDCK cells (1.5 × $10^5$ cells/ml of the collagen gel) were combined with the collagen in DMEM supplemented with 5% FBS, and then polymerized at 37°C in a 5% $CO_2$ atmosphere. The medium was replaced every 2 d. For the DQ-collagen degradation assay, a collagen gel containing 2% DQ-collagen type I (Molecular Probes) was used.

## Mice

Cdcp1-knockout mice were generated in a C57BL/6N using the CRISPR/Cas9 system. Animals were housed in environmentally controlled rooms at the animal experimentation facility at Osaka University. All animal experiments were carried out according to the guidelines of the Osaka University committee for animal and recombinant DNA experiments and were approved by the Osaka University Institutional Review Board. The sequence of gRNA and primers used in genotyping are listed in Table S4 (see also Fig S9A).

## Antibodies and inhibitors

The following primary antibodies were used in this study: anti-CDCP1 (4115), anti-CDCP1 pY734 (9050), anti-STAT3 (9132), anti-STAT3 pY705 (9145), anti-myc-tag (2276), anti-Met pY1234/1235 (3077), anti-Met pY1349 (3121) anti-Met (3127, clone 25H2), and anti-Met (8198) antibodies were all purchased from Cell Signaling Technologies, anti-CDCP1 antibody (LC-C172540) was purchased from LSBio, anti-SFK (sc-18, clone SRC2), anti-ERα (sc-542, clone MC-20), anti-Gapdh (sc-32233, clone 6C5), and anti-MMP2 (sc-10736) antibodies were all purchased from Santa Cruz Biotechnology, anti-Src pY418 (44-660G), anti-Src pY529 (44-662G), and anti-Ki67 (14-5698-82) antibodies were all purchased from Thermo Fisher Scientific, anti-Src (OP07, clone Ab-1), anti-phosphotyrosine (05-1050, clone 4G10), and anti-MMP9 (444236) antibodies were all purchased from Millipore, anti-Fyn (610163, clone 25), anti-Lyn (610003, clone 42), and anti-Yes (610375, clone 1) were all purchased from BD Bioscience, anti-collagen IV antibody (ab6586) was purchased from Abcam, anti-laminin antibody (L9393) was purchased from Sigma-Aldrich, and

---

of hepatocyte growth factor-induced adaptive renal regeneration. CDCP1-Src regulates the Met-STAT3 signaling leading to compensatory renal growth through induction of ECM rearrangement and cell growth/proliferation. Data information: In (A, G), the mean ratios ± SD were obtained from at least six (A) or three mice (G) per group. *P < 0.05; **P < 0.01; ****P < 0.0001; NS, not significantly different; two-way ANOVA was calculated compared with sham-operated control or between wild-type and knockout mice.

anti-keratin 14 antibody (PRB-155P, clone AF64) was purchased from Covance.

The following inhibitors were used in this study: S3i-201 (573102), Stattic (573099), JAK inhibitor 1 (420099), c-Met/Ron dual kinase inhibitor (448104), and Rac1 inhibitor (553502) were purchased from Calbiochem, Marimastat (M2699) was purchased from Sigma-Aldrich, Y27632 (257-00511) from Wako, Saracatinib (AZD0530, S1006) was from Selleck, and other inhibitors listed in Table S1 were all gifted from the Screening Committee of Anticancer Drugs.

## Immunoblotting and immunoprecipitation

For two-dimensional culture, cells were lysed in $n$-octyl-$\beta$-D-glucopyranoside (ODG) buffer (20 mM Tris–HCl [pH 7.4], 150 mM NaCl, 1 mM EDTA, 1 mM $Na_3VO_4$, 20 mM NaF, 1% Nonidet P-40, 5% glycerol, 2% ODG and a protease inhibitor cocktail [Nacalai Tesque]), and immunoblotting was performed. For three-dimensional culture, the cyst-containing collagen matrix was incubated with HBS buffer (10 mM Hepes [pH 7.3], 140 mM NaCl, 4 mM KCl, 1.8 mM $CaCl_2$, and 1 mM $MgCl_2$) containing 0.1% collagenase (Roche) at 37°C. Cysts were harvested by centrifugation and lysed in SDS sample buffer (50 mM Tris–HCl [pH 6.8], 2% SDS, 100 mM NaCl, 1 mM EDTA, 1 mM $Na_3VO_4$, 20 mM NaF and 5% sucrose) before immunoblotting. For immunoprecipitation assays, cells were lysed in ODG buffer and the lysates were incubated with a specific antibody for 1 h at 4°C. Immunoprecipitated proteins were pulled down with protein A- or G-sepharose (GE Healthcare) for immunoblotting. HRP-conjugated anti-mouse or anti-rabbit IgG (Zymed) was used as the secondary antibody. All immunoblots were visualized and quantitated using a Luminograph II System (Atto). Silver staining was performed using the Silver Stain MS Kit (Wako).

## DRM fractionation

Cells were lysed in homogenization buffer (50 mM Tris–HCl [pH 7.4], 150 mM NaCl, 1 mM EDTA, 1 mM $Na_3VO_4$, 20 mM NaF, 0.13–0.25% Triton X-100 and protease inhibitor cocktail) and separated on a discontinuous sucrose gradient (5–35–40%) by ultracentrifugation at 150,000$g$ for 12 h at 4°C using an Optima L-100XP centrifuge equipped with a SW55Ti rotor (Beckman Coulter). The 11 fractions were collected from the top of the sucrose gradient.

## Microarray analysis and quantitative real-time PCR

For microarray analysis, total RNA was isolated from MDCK cysts using the Sepasol-RNA Kit (Nacalai Tesque). Microarray analysis was performed on a G2505C Microarray Scanner (Agilent Technologies) using the Canis (V2) Gene Expression Microarray 4×44K (Agilent Technologies). Microarray data were subjected to gene ontology analysis and upstream regulator analysis using the Ingenuity Pathway Analysis Program (Qiagen).

For quantitative real-time PCR analysis, cDNA was prepared from RNA using the Transcriptor First Strand cDNA Synthesis Kit (Roche) according to the manufacturer's instructions. Real-time PCR was performed on a 7900HT Fast Real-Time PCR System (Applied Biosystems) using the Thunderbird qPCR Mix (Toyobo). Total RNA was normalized to expression of the housekeeping gene GAPDH. The primers used in this analysis are listed in Table S5.

## Immunofluorescent microscopy

For two-dimensional culture, cells were grown on collagen type I–coated coverslips, fixed with 4% paraformaldehyde, and then permeabilized with PBS containing 0.03% Triton X-100. For three-dimensional culture, cysts embedded within the collagen matrix were fixed with 4% paraformaldehyde and permeabilized with PBS containing 0.5% Triton X-100. Permeabilized cells and cysts were blocked with 1% BSA and incubated with primary antibodies for overnight at 4°C, and then incubated with Alexa Fluor 488/594-phalloidin (Molecular Probes) for 2 h at room temperature. For the kidney sections, kidneys were prepared from perfusion fixation with 4% paraformaldehyde and dissected. The fixed kidneys were embedded in OCT compound (Sakura Finetek), sectioned, and mounted on glass slides. The kidney sections were blocked with Blocking One (Nacalai Tesque), incubated with primary antibodies for overnight at 4°C, and then incubated with Alexa Fluor 488/594–conjugated secondary antibodies (Molecular Probes) and the fluorescein-conjugated proximal tubule marker, FITC-LTL (FL-1321; Vector Laboratories), for 2 h at room temperature. Nuclei were counterstained with DAPI. Immunostained objects were observed under a FV1000 confocal microscope (Olympus). For time-lapse observation of three-dimensional cultures, cysts embedded within in collagen matrix were observed under a FV1200 confocal microscope (Olympus).

## Morphometric analyses of MDCK cysts, renal tubules, and glomeruli

Morphometric analyses of MDCK cysts were performed using images data obtained by a FV1000 confocal microscope (Olympus). Phalloidin-stained MDCK cysts were randomly selected, and their morphometric characteristics were measured by using the ImageJ/Fiji software (https://fiji.sc).

Morphometric analyses of kidney were performed as previously described (74, 75). For analysis of renal proximal tubules, kidney sections were stained with LTL-FITC (a specific lectin for renal proximal tubules) and a specific anti-collagen IV antibody. The LTL-positive tubules were visualized under a FV1000 confocal microscope (Olympus). Circular proximal tubules were randomly selected, and their morphometric characteristics were measured by using the ImageJ/Fiji software. Mean thickness of proximal tubules was calculated by 50 circular tubules (five mice/group). Mean area of proximal tubules was calculated by 50 circular tubules (five mice/group). Relative intensity of collagen IV surrounding proximal tubules was calculated by 50 circular tubules (five mice/group). For analysis of glomeruli, the kidney sections were subjected to the HE staining and observed under a BX60 microscope equipped with a DP73 camera (Olympus) or a BZ-X800 microscope (Keyence). Glomeruli were randomly selected, and their morphometric characteristics were measured by using the ImageJ/Fiji software. Mean area was calculated by 50 glomeruli (five mice/group).

## CRISPR/Cas9-based generation of *CDCP1*-knockout MDCK cells

Target-gRNA containing the pSilencer1.0-U6 plasmid, Cas9, and the EGFP co-expressing plasmid (pMJ920; Addgene) were transfected into MDCK cells with MDCK Cell Avalanche Transfection Reagent according to the manufacturer's protocol (EZ Bioscience). 3 d after transfection, EGFP-positive single cells were isolated with a FACSAria III Sorter (BD Biosciences). Knockout of the *Canis CDCP1* gene was confirmed by immunoblotting. The sequence of the gRNA and the primers used in genotyping are listed in Table S4 (see also Fig S3B).

## Plasmid construction and gene transfer

CDCP1, CDCP1 deletion mutants, STAT3, and Met were generated by PCR using human cDNA as the template and subcloned into the pCX4 retroviral plasmid (generously donated by Dr. Akagi) (76). CDCP1 mutants (K365A-R368A-K369A, C689G-C690G, Y762F, and Y734F) and STAT3-Y705F were generated by mutagenesis PCR using KOD-Plus polymerase (Toyobo). CDCP1 and its respective mutants were subcloned into either pEGFP-N1 or pmCherry-N1 plasmid (Clontech) and then further subcloned into the pRetroX-TRE3G retroviral plasmid (Clontech). Src-MER and STAT3-MER were constructed by modifying ligand-binding domain of the estrogen receptor (MER, amino acids 281–599) and subcloning into the pCX4 plasmid (27). mCherry-CAAX was constructed using the C-terminal region of human KRAS (amino acids 166–189) and subcloning into the pCX4 plasmid. mCherry-GPI was also subcloned into the pCX4 plasmid (generously donated by Dr. Kiyokawa) (77). All constructs were confirmed by sequencing. Gene transfer of pCX4 and pRertroX-TRE3G was carried out by retroviral infection. Retroviral production and infection were performed as described previously (78).

## Surgical procedures of mice

Compensatory renal growth was induced by right UNX in male mice (8 wk of age) under anesthesia as previously described (3). Renal growth was evaluated by measuring the weight of remaining kidney and the body. Left kidneys of sham-nephrectomized mice were used as controls for UNX mice.

## Statistics and reproducibility

For data analyses, unpaired two-tailed *t* tests were used to determine the *P*-values. For multiple group comparisons, ANOVA was used. A *P*-value less than 0.05 was considered to be significant. All data and statistics were derived from at least three independent experiments.

## Data Availability

Supporting microarray data have been deposited in the Gene Expression Omnibus under accession code GSE99375. All other supporting data are available from the corresponding author upon request.

# Supplementary Information

# Acknowledgements

We thank Dr. Takata for consultation on experiments, Dr. Kiyokawa for providing the mCherry-GPI plasmid, Dr. Saito for mass spectrometry analysis, and the Spectrography and Bioimaging Facility (NIBB Core Research Facilities). The inhibitor kit was provided by the Screening Committee of Anticancer Drugs supported by Grant-in-Aid for Scientific Research on Innovative Areas, Scientific Support Programs for Cancer Research, from The Ministry of Education, Culture, Sports, Science and Technology of Japan. This work was supported by Grant-in-Aid for Young Scientists (B) (26830071, to K Kajiwara), Grant-in-Aid for Scientific Research (C) (19K07639, to K Kajiwara), Grant-in-Aid for Scientific Research (B) (19H03504, to M Okada), Grant-in-Aid for Scientific Research on Innovative Area (19H04962, to M Okada), Grant-in-Aid for Scientific Research on Innovative Area (16H01447, to K Aoki) from the Ministry of Education, Culture, Sports and Technology of Japan; the Takeda Science Foundation to K Kajiwara; and the Extramural Collaborative Research Grant of Cancer Research Institute, Kanazawa University to K Kajiwara.

## Author Contributions

K Kajiwara: conceptualization, resources, data curation, formal analysis, funding acquisition, investigation, visualization, methodology, project administration, and writing—original draft, review, and editing.
S Yamano: validation and investigation.
K Aoki: funding acquisition, investigation, and visualization.
D Okuzaki: data curation, formal analysis, and investigation.
K Matsumoto: resources and supervision.
M Okada: conceptualization, resources, supervision, funding acquisition, project administration, and writing—review and editing.

## Conflict of Interest Statement

The authors declare that they have no conflict of interest.

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
