## [Reviewer comments · Life Science Alliance]

Life Science Alliance

CDCP1 promotes compensatory renal growth by integrating Src and Met signaling

Kentaro Kajiwara, Shotaro Yamano, Kazuhiro Aoki, Daisuke Okuzaki, Kunio Matsumoto, and Masato Okada

DOI: <https://doi.org/10.26508/lsa.202000832>

Corresponding author(s): Kentaro Kajiwara, Research Institute for Microbial Diseases, Osaka University

Review Timeline:

Submission Date:	2020-06-26
Editorial Decision:	2020-07-30
Revision Received:	2020-12-03
Editorial Decision:	2021-01-12
Revision Received:	2021-01-19
Accepted:	2021-01-19

Scientific Editor: Shachi Bhatt

Transaction Report:

July 30, 2020

Re: Life Science Alliance manuscript #LSA-2020-00832-T

Author information redacted

Dear Dr. Kajiwara,

Thank you for submitting your manuscript entitled "CDCP1 promotes compensatory renal growth by integrating Src and Met signaling" to Life Science Alliance. The manuscript was assessed by expert reviewers, whose comments are appended to this letter.

As you can see, while the referees find your work potentially interesting, they also raise major points that need to be addressed before they can support publication in Life Science Alliance. In particular, referee #1 asks you to use Src-/- MDCK cells to test that the pY418 staining at the tip of cell protrusion is due to the recruitment of c-Src rather than Yes or Fyn kinases and to confirm the results obtained with dasatinib. In addition, reviewer #2 finds that the two main parts of the study (i.e. MDCK system and in vitro analysis and CdcP1 mouse model) need to be better integrated. Also, s/he recommends performing Ki67 quantification of cysts in vitro and to further investigate the tubular thickening phenotype of the UNX model.

We agree with the referees that these are important points and addressing these and the other referees' questions would be essential to pursue publication of this study in Life Science Alliance. In addition, we will need strong support from the referees for publication here.

Given the overall interest of your study, we would like to invite you to submit a revised version of the manuscript according to the referee's requests. We should add that it is Life Science Alliance's policy to allow only a single round of revision, and acceptance of your manuscript will therefore depend on the completeness of your responses in this revised version.

We realize that addressing all the referees' criticisms may require a lot of additional time and effort and be technically challenging. We would therefore understand if you were to choose not to undergo an extensive revision here and rather pursue a submission at an alternative venue, in which case please inform us about your decision at your earliest convenience.

In our view these revisions should typically be achievable in around 3 months. However, we are aware that many laboratories cannot function fully during the current COVID-19/SARS-CoV-2 pandemic and therefore encourage you to take the time necessary to revise the manuscript to the extent requested above. We will extend our 'scoping protection policy' to the full revision period required. If you do see another paper with related content published elsewhere, nonetheless contact me immediately so that we can discuss the best way to proceed.

Thank you for this interesting contribution to Life Science Alliance. We are looking forward to receiving your revised manuscript.

Sincerely,

Reilly Lorenz
Editorial Office Life Science Alliance
Meyerhofstr. 1
69117 Heidelberg, Germany
t +49 6221 8891 414
e contact@life-science-alliance.org
www.life-science-alliance.org

B. MANUSCRIPT ORGANIZATION AND FORMATTING:

Reviewer #1 (Comments to the Authors (Required)):

Here the authors have investigated the function of the CDCP1/Trask transmembrane adaptor protein in HGF/HGF receptor (Met) tyrosine kinase signaling in cultured MDCK cell cysts, used as a model of renal tubule formation, a process that is important in compensatory renal growth which requires HGF signaling. They started by showing that HGF treatment induced protrusions in these cyst structures, and that their outgrowth depended on mTORC1 signaling and was inhibited by dasatinib, a Src/Abl kinase inhibitor. They found that the tips of the HGF-induced protrusions were stained with anti-pY418 Src antibodies, indicative of activated Src, and also marked by mCherry-GPI, a lipid raft reporter. Activation of stably expressed Src-MER with 4-OHT also induced formation of protrusions, further implicating Src activity in this morphological process. Since the Src protein is not normally localized to lipid rafts, they focused on CDCP1, a palmitoylated, transmembrane scaffolding protein that is known to be lipid raft localized and when tyrosine phosphorylated binds Src via its SH2 domain. Consistent with a role for CDCP1, they found that CDCP1 was transiently located at the tips of protrusions upon HGF treatment, and that *Cdcp1*^{-/-} MDCK cells did not exhibit protrusions when treated with HGF. They also showed that Dox-induced overexpression of CDCP1 in MDCK cells increased the level of pY418 Src in the absence of HGF, whereas expression of a Y734F mutant form of CDCP1 that cannot bind Src did not. Next, they showed that expression of WT CDCP1 in MDCK cells increased pY705 STAT3 levels and downstream STAT3 transcriptional signaling, with WT CDCP1 being more effective than overexpressed CDCP1 C689/690G palmitoylation site mutant. WT CDCP1 expression led to increased expression of several genes, including MMPs, and MMP expression in turn led to disruption of the basement membrane and the extension of protrusions, which was suppressed by the marimastat MMP inhibitor. Consistent with their model, activation of STAT3-MER with 4-OHT led to the same phenotypes, and the effect of CDCP1 overexpression was blocked by small molecule STAT3 inhibitors. In investigating the possible link between CDCP1 and HGF-MET, they showed that CDCP1 and Met co-precipitated, and that deletion of the CDCP1 N-terminal CUB domain increased this interaction, implying that CUB1 is a negative regulator of Met binding, whereas the other two CDCP1 CUB domains were required for Met binding. Since CDCP1 is physiologically cleaved between the first two CUB domains, the authors deduced that it was the cleaved form of CDCP1 that binds Met. They showed that expression of a non-cleavable mutant form of CDCP1 activated Src, but failed to bind Met, and on this basis proposed that cleaved CDCP1/Src complexes localized in lipid rafts promote Met-mediated activation of STAT3 phosphorylation and signaling. Finally, they went on to assess whether this pathway might play a role in compensatory renal growth in vivo, and found that the proximal renal tubule expansion observed following unilateral nephrectomy (UNX) was reduced in *Cdcp1*^{-/-} mice. Consistent with their model, the observed induction of pY1234/5 Met and pY705 STAT3 in the remaining kidney was decreased in *Cdcp1*^{-/-} UNX mice, concomitant with a lack of Ki67⁺ proliferating cells. In contrast, CDCP1 pY734 levels were increased in the remaining kidney in WT UNX mice, as was the number of Ki67⁺ cells, and expression of MMP2 and MMP9.

This is a nice paper with evidence that the CDCP1/Trask transmembrane scaffolding protein in conjunction with the Src tyrosine kinase plays a role in HGF/Met induction of tubule formation in

MDCK kidney epithelial cell spheroids and in compensatory renal growth in unilaterally nephrectomized mice. The interaction between cleaved CDCP1 and Met is a new finding, and could explain the HGF-induced recruitment of Src to Met and the phosphorylation of STAT3 at Y705, which is important for the outgrowth of protrusions and ultimately for compensatory kidney growth. There are a number of questions.

1. In the first part of the paper the authors focused on a role for the c-Src SFK, which is not normally localized to lipid rafts, in the HGF induced outgrowth of MDCK cyst protrusions. However, they did not actually show that the pY418 signal they observed at the tips of protrusions is in fact due to recruitment c-Src itself, as opposed to recruitment of the Yes and Fyn SFKs; Yes and Fyn are naturally localized to lipid rafts, because they possess one or two palmitoylated Cys residues, respectively, within their N-terminal unique sequences (aa1-15). Moreover, both the anti-pY418 Src and the anti-pY529 Src antibodies used here will recognize the equivalent activating and inhibitory pTyr residues in Fyn and Yes, because the sequences around them are identical to those in c-Src. In addition, the Fyn and Yes SH2 domains will also bind to pY734 in CDCP1, especially when it is localized in lipid rafts. The authors used dasatinib, a Src/Abl inhibitor to establish a role for Src in HGF-induced protrusions, but dasatinib will also inhibit Yes and Fyn equally well (see below). Further evidence is needed that Src is the central player in the proposed pathway (e.g. using Src knockout MDCK cells).

2. Did the authors establish whether Met/CDCP1 coprecipitation requires CDCP1 palmitoylation? This could be checked using the CDCP1 C689/690G palmitoylation mutant. This gets at the question of whether Met also translocates to lipid rafts with CDCP1. No Met staining or "Met-GFP" expressing cells experiments are presented to determine this. It is also unclear whether HGF treatment increases Met/CDCP1 association/co-precipitation. In this regard, it would be important to show that association can be detected between endogenous Met and endogenous CDCP1 in response to HGF treatment.

3. The authors showed that Met kinase activity is required for CDCP1 overexpression to induce protrusions, and presumably it is also required for the HGF-induced protrusions, although they did not show this. Is there a Met^{-/-} MDCK line that could be used to establish this another way?

4. In their model the authors propose that STAT3 Y705 is phosphorylated by Src, which is known to phosphorylate Y705, following binding of CDCP1/Src to Met/STAT3 in this system? What is unclear is how unphosphorylated STAT3 binds to Met, and whether this requires Met activation, and interaction with a specific Met autophosphorylation site.

5. Does CDCP1 overexpression itself promote Met activation/autophosphorylation in the absence of HGF, and if so how, and is Met autophosphorylation required for STAT3 association in the absence of HGF?

6. Which tyrosine kinase phosphorylates CDCP1 in this system - this phosphorylation obviously has to occur before Src can bind to CDCP1 and be recruited to lipid rafts. Is CDCP1 tyrosine phosphorylation induced by HGF? Is Src itself involved? There are no blots showing CDCP1 Tyr phosphorylation in the MDCK system, and these would strengthen the paper (are good anti-pY734 antibodies available and could they be used to monitor CDCP1 phosphorylation and the localization of pY734 CDCP1 protein in HGF-treated MDCK cysts?).

7. Does the position of a lipid raft in the plasma membrane dictate where a tubule can initiate protrusion in an HGF-treated MDCK cell? The data showing mCherry-GPI at the tips of protrusions

suggests that the tip of the tubule constitutes a lipid raft. Can a lipid raft, which has high cholesterol content, achieve the curvature needed at the tip of a tubule? Otherwise, why is lipid raft localization of CDCP1/Src important for the initiation of tubule outgrowth? Do Met and STAT3 also translocate to the tips of tubules?

8. EGFR is known to interact with CDCP1 - is it clear that EGFR is not also involved in HGF-induced formation of protrusions?

There are also some other issues:

1. Dasatinib is a very "dirty" kinase inhibitor, inhibiting many Tyr (and Ser/Thr) kinases, including the collagen-activated DDR1/2 RTKs (since the MDCK cysts are cultured in collagen, this could be an issue). Moreover, while dasatinib does inhibit c-Src, it inhibits the Yes and Fyn SFKs equally well, and also inhibits Abl (and Arg/Abl2) just as potently. This is important, because Abl is known to induce formation of cellular protrusions when activated (e.g. Woodring et al., JCB 156:879, 2002). Thus, the use of dasatinib as a Src inhibitor is not advised, since any observed phenotypes cannot necessarily be attributed to SFK inhibition. Significantly more selective SFK inhibitors, such as sarcatinib, are available. If the authors want to establish a role for Src per se, they should generate Src^{-/-} MDCK cells.

2. Figure S1A: The authors need to remind the readers of the structure of the Src-MER protein, which they developed previously, i.e. the MER domain is appended to the C-terminus, and indicate whether the presence of the C-terminal MER domain affects negative regulation of Src activity by Csk phosphorylation.

3. Figure 2C: It appears that there is residual CDCP1 in the CDCP1KO#1 lane - does this mean that these Cdcp1^{-/-} MDCK cells were not clonal?

4. Is there a role for any of the other pTyr sites in CDCP1 in this pathway, and is their phosphorylation induced by HGF treatment?

Reviewer #2 (Comments to the Authors (Required)):

In this study by Kajiwara, et al entitled "CDCP1 promotes compensatory renal growth by integrating Src and Met signalling" the authors present a novel signalling axis involving the scaffolding protein CDCP1, HGF and STAT3 signaling, which controls epithelial cell proliferation and ECM remodelling. Moreover, the authors integrate this signalling network in the context of compensatory organ growth and employ unilateral nephrectomy in the mouse as a model system.

In general, the study is employing a wide variety of different methodologies including complex 3D culture systems, genome engineering, proteomics and biochemistry approaches as well as in vivo mouse models. The majority of experiments are performed in a very stringent and detailed fashion, accompanied by a clear and easy-to-follow description in the manuscript.

The manuscript is clearly separated into two different aspects, where one is exclusively using MDCK cells in 3D as a model system for renal tubular epithelial cells and the other one is analysing the Cdcp1 mouse model. Here, each part for its own is very elegantly presented (the in vitro part in much more detail), but the synthesis, combination and conclusions from both parts are less convincing than the individual aspects. Based on that the authors should be careful in the interpretation of the data and conclusions.

1) MDCK system and in vitro analysis:

This part of the manuscript is really extensive and provides interesting novel insights about Cdc42 function in the MDCK cyst model. Genetic as well as inhibitor studies are elegantly performed.

Following points should be addressed:

-the authors use a wide variety of chemical compounds and inhibitors (e.g. figure 1) - here they state that mTOR inhibition suppresses cell proliferation of cysts. The presented IF images also illustrate a dramatically altered morphology of individual cysts. The authors should also discuss effects on the cytoskeleton or viability of the cells due to compound treatments. Also the statement that dasatinib suppresses proliferation is definitely true looking at the cyst diameters, however at the same time Ki67 positivity is rather unaffected in the presented images. Maybe Ki67 quantification of cysts could clarify the impact on proliferation and could exclude secondary effects on cyst morphology due to cytoskeleton rearrangement or increased cell death (or cell volume regulation?).

-the authors argue that CDC42 is tethered to lipid rafts (this is also based on biochemistry experiments). In Figure 1 the authors use mCherry-GPI sensors to demonstrate the localization of CDC42 towards lipid rafts, but the IF stainings show more or less exclusive localization at cellular protrusions (would a co-staining approach be feasible to demonstrate a clear co-localization for the protein and the sensor)? This would help to clarify the exact localization pattern of CDC42 in 3D culture.

-based on the cellular phenotype and also the microarray data the authors identify ECM remodelling as a central downstream effect of CDC42 function (also involving MMP signalling). This is also elegantly investigated using genetic interference and chemical compounds. In the discussion the authors mention that CDC42 is also promoting this kind of phenotype in cancer tissues, however, it is less clear how this "invasive" phenotype might reflect any aspect of kidney tubule function? ECM degradation (as required for this invasive behaviour) is not required for tubular regeneration or growth. Based on that, it is highly questionable whether this phenotype really reflects kidney epithelial cell behaviour in a meaningful way. The reviewer is well aware that this model (incl. HGF application) is widely used and accepted, but nevertheless the authors should clearly state that this is only used as a proxy to model CDC42 effects in epithelial cells (see also below). The transfer of the observed phenotype to the in vivo condition is clearly a difficult point in the whole manuscript.

2) Cdc42 mouse model:

-the tubular thickening phenotype described in the UNX model is really questionable - at least on the presented images the difference between WT and KO mice is not easily to distinguish. Has this kind of quantification been used before in other publications? Is this approach validated at different time points? Such kind of measurements are really prone for artefacts and therefore a clear description of sample preparation and measurement and quantification approach is required (also HE/PAS stains of the kidneys should be presented, preferably whole slide scans and close ups).

-what is the absolute effect on the kidney weight? Only kidney/body weight ratios are presented making it difficult to grasp the absolute effect on the kidney mass. While the difference in ratios between WT/KO mice is significant, the delta of the ratio is rather modest. This makes it a little bit questionable whether knockout mice really have a dramatic deficit in organ growth. These points should be addressed by the authors (this holds also true for the proliferation rate - here is only a difference of about 2% observed in the Ki67 proliferation rate; whether that might really relate to impaired organ growth needs to be discussed).

-is there any other published evidence of collagen IV breakdown in the tubular basement membrane (TBM) in conditions of UNX or tubular damage? The TBM is usually a very stable structure and dissolution is only observed in degenerating tubular compartments. As the authors try to connect the ECM/invasion phenotype from their in vitro model with the in vivo situation, the

authors should (if they really want to emphasize this point) try to quantitate these effects and corroborate those findings by ultrastructural techniques. Based on published work this aspect of organ growth and TBM remodelling is not really established (also the cited reference "34" has a complete different focus).

-minor: IF stainings should be quantified, where differences have been observed.

Reviewer #3 (Comments to the Authors (Required)):

Kajiwara et al. identify a function of CDCP1 in regulating changes in morphology and growth of renal cysts. CDCP1 expression is required and sufficient for formation of protrusions in MDCK cell cysts; moreover, CDCP1 deficiency prevents renal cyst growth in an in vivo mouse model. The authors confirm a role for CDCP1 in Src - STAT3 signaling to mediate these changes and provide evidence that these signaling events require lipid raft localization of CDCP1-Src. The present study extends known functions of CDCP1 in signal transduction by interesting observations about the importance of subcellular localization and the implication for renal growth. The experiments are of overall high quality and support the author's conclusions. The data are clearly presented in a well-written manuscript, and I recommend this work for publication in Life Science Alliance. A few suggestions to further improve the manuscript are outlined below.

1. The authors propose in their model (Fig. 6H) that CDCP1-Src specifically upregulate Stat3 downstream of Met, suggesting the interesting possibility that CDCP1 fine-tunes Met signaling by selective activation of some downstream effectors. Consistently, I have the impression that Dasatinib potently suppresses formation of cyst protrusions but is less effective than Torin 1 or Rapamycin in suppressing cell proliferation (Fig. 1A-C). It would be interesting to know whether CDCP1 overexpression / inhibition influences mTORC1 activity in HGF-stimulated cells.
2. Fig. 1D, F: If possible, it would be good to co-stain for p-Src and CDCP1 in the same sample.
3. Suppl. Fig. 1C: Do the numbers on the left side indicate time? If yes, which unit?
4. Suppl. Fig. 7B: Which biological samples were used for the PCR? A Western blot of CDPC1 in wt / KO would be helpful.
5. Suppl. Fig. 8A: It is a bit difficult to tell which p-CDCP1 staining is specific vs non-specific (compare e.g. Sham Cdcp1 -/- and UNX 0.5 d Cdcp1 +/+).
6. In the methods section, please give the antibody incubation times for immunoprecipitation and immunostainings.

Point-by-point response letter

LSA-2020-00832-T

CDCP1 promotes compensatory renal growth by integrating Src and Met signaling

Kentarō Kajiwara, Shotaro Yamano, Kazuhiro Aoki, Daisuke Okuzaki, Kunio Matsumoto and Masato Okada

Reviewer #1 (Comments to the Authors (Required)):

Here the authors have investigated the function of the CDCP1/Trask transmembrane adaptor protein in HGF/HGF receptor (Met) tyrosine kinase signaling in cultured MDCK cell cysts, used as a model of renal tubule formation, a process that is important in compensatory renal growth which requires HGF signaling. They started by showing that HGF treatment induced protrusions in these cyst structures, and that their outgrowth depended on mTORC1 signaling and was inhibited by dasatinib, a Src/Abl kinase inhibitor. They found that the tips of the HGF-induced protrusions were stained with anti-pY418 Src antibodies, indicative of activated Src, and also marked by mCherry-GPI, a lipid raft reporter. Activation of stably expressed Src-MER with 4-OHT also induced formation of protrusions, further implicating Src activity in this morphological process. Since the Src protein is not normally localized to lipid rafts, they focused on CDCP1, a palmitoylated, transmembrane scaffolding protein that is known to be lipid raft localized and when tyrosine phosphorylated binds Src via its SH2 domain. Consistent with a role for CDCP1, they found that CDCP1 was transiently located at the tips of protrusions upon HGF treatment, and that *Cdcp1*^{-/-} MDCK cells did not exhibit protrusions when treated with HGF. They also showed that Dox-induced overexpression of CDCP1 in MDCK cells increased the level of pY418 Src in the absence of HGF, whereas expression of a Y734F mutant form of CDCP1 that cannot bind Src did not. Next, they showed that expression of WT CDCP1 in MDCK cells increased pY705 STAT3 levels and downstream STAT3 transcriptional signaling, with WT CDCP1 being more effective than overexpressed CDCP1 C689/690G palmitoylation site mutant. WT CDCP1 expression led to increased expression of several genes, including MMPs, and MMP expression in turn led to disruption of the basement membrane and the extension of protrusions, which was suppressed by the marimastat MMP inhibitor. Consistent with their model, activation of STAT3-MER with 4-OHT led to the same phenotypes, and the effect of CDCP1 overexpression was blocked by small molecule STAT3 inhibitors. In investigating the possible link between CDCP1 and HGF-MET, they showed that CDCP1 and Met co-precipitated, and that deletion of the CDCP1 N-terminal CUB domain increased this interaction, implying that CUB1 is a negative regulator of Met binding, whereas the other two CDCP1 CUB domains were required for Met binding. Since CDCP1 is physiologically cleaved between the first two CUB domains, the authors deduced that it was the cleaved form of CDCP1 that binds Met. They showed that expression of a non-cleavable mutant form of

CDCP1 activated Src, but failed to bind Met, and on this basis proposed that cleaved CDCP1/Src complexes localized in lipid rafts promote Met-mediated activation of STAT3 phosphorylation and signaling. Finally, they went on to assess whether this pathway might play a role in compensatory renal growth in vivo, and found that the proximal renal tubule expansion observed following unilateral nephrectomy (UNX) was reduced in *Cdcp1*^{-/-} mice. Consistent with their model, the observed induction of pY1234/5 Met and pY705 STAT3 in the remaining kidney was decreased in *Cdcp1*^{-/-} UNX mice, concomitant with a lack of Ki67+ proliferating cells. In contrast, CDCP1 pY734 levels were increased in the remaining kidney in WT UNX mice, as was the number of Ki67+ cells, and expression of MMP2 and MMP9.

This is a nice paper with evidence that the CDCP1/Trask transmembrane scaffolding protein in conjunction with the Src tyrosine kinase plays a role in HGF/Met induction of tubule formation in MDCK kidney epithelial cell spheroids and in compensatory renal growth in unilaterally nephrectomized mice. The interaction between cleaved CDCP1 and Met is a new finding, and could explain the HGF-induced recruitment of Src to Met and the phosphorylation of STAT3 at Y705, which is important for the outgrowth of protrusions and ultimately for compensatory kidney growth. There are a number of questions.

1. In the first part of the paper the authors focused on a role for the c-Src SFK, which is not normally localized to lipid rafts, in the HGF induced outgrowth of MDCK cyst protrusions. However, they did not actually show that the pY418 signal they observed at the tips of protrusions is in fact due to recruitment c-Src itself, as opposed to recruitment of the Yes and Fyn SFKs; Yes and Fyn are naturally localized to lipid rafts, because they possess one or two palmitoylated Cys residues, respectively, within their N-terminal unique sequences (aa1-15). Moreover, both the anti-pY418 Src and the anti-pY529 Src antibodies used here will recognize the equivalent activating and inhibitory pTyr residues in Fyn and Yes, because the sequences around them are identical to those in c-Src. In addition, the Fyn and Yes SH2 domains will also bind to pY734 in CDCP1, especially when it is localized in lipid rafts. The authors used dasatinib, a Src/Abl inhibitor to establish a role for Src in HGF-induced protrusions, but dasatinib will also inhibit Yes and Fyn equally well (see below). Further evidence is needed that Src is the central player in the proposed pathway (e.g. using Src knockout MDCK cells).

Response #1

Based on this suggestion, we examined the contributions of Src family kinases (SFKs) other than Src. As shown in Figure 3A, 4D, 5D, S4A and S4C, multiple bands were visible in the Src pY418 blot following CDCP1 overexpression, indicating that activation of multiple SFK proteins was involved in CDCP1-mediated signaling. In fact, previous studies have reported that CDCP1 associates with Src, Fyn, and Yes (Bhatt, *Oncogene*, 2005; Uekita, *Mol Cell Biol*, 2007; Wong, *Clin Cancer Res*, 2009). RNA-Seq analysis of MDCK

cells detected the expression of five SFKs (Lyn, Yes, Src, Fyn, and Hck) (see Fig. 1 for review), and mass spectrometric analysis of the lipid raft (DRM) fractions detected Lyn and Yes, in addition to Src (**Fig. S4E**). Following CDCP1 overexpression, Lyn and Yes relocated into the DRM fractions (**Fig. S4F and G**) and phosphorylated Lyn and Yes were associated with CDCP1 in a manner similar to Src (**Fig. S4D**). These results indicate that CDCP1 controls multiple SFKs in MDCK cells. Therefore, to evaluate the roles of SFKs, we chose to use the broad-spectrum SFK inhibitor, dasatinib, instead of Src knockout cells. In accordance with the reviewer's suggestion, we used a more specific SFK inhibitor, saracatinib, to ensure the roles of SFKs (**Fig. 1A–D, 2D, 2E, 5B–F, and Fig. S8E–G**). In this study, however, we have chosen to mainly focus on Src as a representative SFK.

[Figure removed by editorial staff per authors' request]

2. Did the authors establish whether Met/CDCP1 coprecipitation requires CDCP1 palmitoylation? This could be checked using the CDCP1 C689/690G palmitoylation mutant. This gets at the question of whether Met also translocates to lipid rafts with CDCP1. No Met staining or "Met-GFP" expressing cells experiments are presented to determine this.

Response #2

We thank the reviewer for raising this question. We observed that palmitoylation-deficient CDCP1 (CDCP1-C689/690G) associates with Met (see Fig. 2A for review). However, wild-type CDCP1 recruited Met to the lipid rafts, whereas CDCP1-C689/690G did not (**Fig. S8D**). These observations indicate that CDCP1 palmitoylation is required to co-translocate Met to lipid rafts. Furthermore, STAT3 activation was substantially attenuated in CDCP1-CG overexpressing cells (see Fig. 2B for review and **Fig. 3A**). Taken together, these data suggest that CDCP1-Met interaction in lipid rafts is required for efficient activation of downstream signaling pathways.

[Figure removed by editorial staff per authors' request]

It is also unclear whether HGF treatment increases Met/CDCP1 association/co-precipitation. In this regard, it would be important to show that association can be detected between endogenous Met and endogenous CDCP1 in response to HGF treatment.

Response #3

In accordance with the second comment, we analyzed the association between endogenous CDCP1 and Met proteins using two renal cancer cell lines, A498 and ACHN, with high endogenous levels of CDCP1 and Met (**Fig. S7B**). Immunoprecipitation analyses showed that the two proteins associated with each other, and these interactions were not affected by HGF treatment. Interestingly, CDCP1 recruits and traps Met on the plasma membrane, and this event was not inhibited by treatment with the HGF antagonist, NK4 (**Fig. 5B and C**). These findings suggest that CDCP1 associates with Met independently of HGF stimulation.

3. The authors showed that Met kinase activity is required for CDCP1 overexpression to induce protrusions, and presumably it is also required for the HGF-induced protrusions, although they did not show this. Is there a Met^{-/-} MDCK line that could be used to establish this another way?

Response #4

We could not generate the Met knockout MDCK cells using a CRISPR/Cas9 system. Indeed, only one study using Met knockout cell lines has been reported (Miao, *Cancer Sci*, 2019). To assess the requirement of Met kinase activity in other way, we analyzed the HGF-induced protrusion with Met kinase inhibitors,

PHA665752 and Met/Ron dual kinase inhibitor (**Fig. S1A–C**). For the same reason, we used Met kinase inhibitors to analyze CDCP1-mediated morphological changes (**Fig. 4A and Table S1**). These results indicate that Met kinase activity is required for the HGF-induced protrusion formation.

4. In their model the authors propose that STAT3 Y705 is phosphorylated by Src, which is known to phosphorylate Y705, following binding of CDCP1/Src to Met/STAT3 in this system? What is unclear is how unphosphorylated STAT3 binds to Met, and whether this requires Met activation, and interaction with a specific Met autophosphorylation site.

Response #5

This comment is quite insightful and helpful for improving this study. We analyzed Met phosphorylation in CDCP1-overexpressing MDCK cells. Following CDCP1 overexpression, both the expression level of endogenous Met and the phosphorylation at its kinase domain (Y1234/1235) was increased (**Fig. 5A and Fig. S8B**). Furthermore, subsequent phosphorylation at the autophosphorylation site (Y1349) that is required for association with STAT3 through its SH2 domain (Boccaccio, *Nature*, 1998), was detected (**Fig. 5A and Fig. S8C**). Under the same conditions, immunoprecipitation analysis showed that phosphorylated Met was associated with STAT3, but not with phosphorylated STAT3 (pY705) (**Fig. 5E and F**). In contrast, these phenomena were suppressed by the Y734F point mutant of CDCP1 or saracatinib treatment (**Fig. 5A, D, E, and F**). These results suggest that Met phosphorylation is induced by CDCP1-associated Src and that this phosphorylation is required for the Met-STAT3 association. Based on these data, we have proposed a new mechanistic model in the revised version of the manuscript (**Fig. 5G**).

5. Does CDCP1 overexpression itself promote Met activation/autophosphorylation in the absence of HGF, and if so how?

Response #6

Thank you for the valuable comment. We have addressed this point by examining the effects of CDCP1 overexpression on Met activity. The results showed that overexpressed CDCP1 recruited endogenous Met protein to the plasma membrane (**Fig. 5B and C**), especially to the lipid raft fraction (**Fig. S8D**), and Met was phosphorylated at both Y1234/1235 and Y1349 (**Fig. 5A and D and Fig. S8I**). This phenomenon was suppressed by point mutations at the Src association site (Y734F) and cleavage site (PR) in CDCP1 (**Fig. 5A**), indicating that accumulation of activated Met requires CDCP1-Src and CDCP1-Met interactions. Because Met is phosphorylated by Src kinase (Emaduddin, *PNAS*, 2008; Rankin, *PNAS*, 2014), Met may be directly phosphorylated by CDCP1-associated Src. However, this phosphorylation was not completely repressed by saracatinib treatment, implying that other Met phosphorylation mechanisms also exist. Furthermore, Met

phosphorylation was not inhibited by treatment with the HGF antagonist, NK4 (**Fig. 5D**), indicating that Met activation was not caused by HGF secreted in the medium. Met is phosphorylated by other growth factor receptors, including EGFR and HER2 that can interact with CDCP1 (Alajati, *Cell Rep*, 2015; Law, *Breast Cancer Res*, 2016). In our experimental conditions, FBS depletion caused downregulation of Met protein and its phosphorylation (**Fig. S8I–K**). These results suggest that CDCP1 functions to trap and/or stabilize Met, allowing Met activation by some membrane tyrosine kinases, potentially including Src. These new findings are presented in Figure 5 and Figure S8, and the molecular mechanism is explained in detail in the Discussion section.

Is Met autophosphorylation required for STAT3 association in the absence of HGF?

Response #7

Previously, it has reported that Met-STAT3 association requires Met autophosphorylation (pY1349) (Boccaccio, *Nature*, 1998). We observed that overexpressed CDCP1 induce Met autophosphorylation (pY1349) even in the absence of HGF stimulation (**Fig. 5A and D and Fig. S8I**). In the same conditions, immunoprecipitation analysis showed that phosphorylated Met interacted with STAT3, but not with phosphorylated STAT3 (pY705) (**Fig. 5E and F**). Furthermore, the Met-STAT3 association was suppressed by treatment with a Met/Ron kinase inhibitor. These data indicate that Met autophosphorylation is required for STAT3 association.

6. Which tyrosine kinase phosphorylates CDCP1 in this system - this phosphorylation obviously has to occur before Src can bind to CDCP1 and be recruited to lipid rafts. Is CDCP1 tyrosine phosphorylation induced by HGF? Is Src itself involved?

Response #8

CDCP1 phosphorylation at Tyr734 was not suppressed by the HGF antagonist, NK4, whereas the Src inhibitor saracatinib suppress this phosphorylation (**Fig. 5D**). These data suggest that CDCP1 was directly phosphorylated by Src. We assume that Src initially phosphorylates CDCP1 at Tyr734 and then bind to CDCP1 to be activated, leading to an amplification of the phosphorylation of CDCP1 via a positive feedback loop.

There are no blots showing CDCP1 Tyr phosphorylation in the MDCK system, and these would strengthen the paper (are good anti-pY734 antibodies available and could they be used to monitor CDCP1 phosphorylation and the localization of pY734 CDCP1 protein in HGF-treated MDCK cysts?).

Response #9

Following the reviewer's suggestion, data showing CDCP1 pY734 blots were depicted on Figure 5E and Figure S4A. However, because anti-CDCP1 pY734 antibody (CST, #9050) was not suitable for immunofluorescence microscopic analyses of two- and three-dimensional culture of MDCK cells (see Fig. 3 for review), we could not visualize CDCP1 pY734 in our experimental system.

[Figure removed by editorial staff per authors' request]

7. Does the position of a lipid raft in the plasma membrane dictate where a tubule can initiate protrusion in an HGF-treated MDCK cell? The data showing mCherry-GPI at the tips of protrusions suggests that the tip of the tubule constitutes a lipid raft. Can a lipid raft, which has high cholesterol content, achieve the curvature needed at the tip of a tubule?

Response #10

Thank you for your valuable comment on lipid rafts. We observed that HGF-induced protrusion formation was inhibited by cholesterol depletion caused by pretreatment with simvastatin (**Fig. 1A and D**). Pretreatment with simvastatin also inhibited HGF-induced cell growth and proliferation (**Fig. 1A–C**). Furthermore, a

previous study reported that HGF-Met signaling is downregulated by disruption of lipid rafts (Gutierrez et al., *Skeletal Muscle*, 2014). These observations underscore the important role of lipid rafts as a membrane platform for HGF-Met signaling. Thus, we believe that lipid rafts achieve curvature at the tip of a tubule, although the underlying molecular mechanisms need to be clarified through future studies.

Otherwise, why is lipid raft localization of CDCP1/Src important for the initiation of tubule outgrowth?

Response #11

We have shown that tubule outgrowth was largely repressed in cells overexpressing lipid raft-mislocalized CDCP1 (CDCP1-CG-EGFP) (**Fig. 2F and G**). In these cells, STAT3 activation, which was required for CDCP1-mediated responses, was substantially weaker than that in wild-type CDCP1-overexpressing cells (**Fig. 3A**). Therefore, it is likely that lipid raft localization of CDCP1-Src is important for the specific activation of STAT3 in lipid rafts, which leads to tubule outgrowth associated with MMP production. We have described this point in detail in the revised manuscript.

Do Met and STAT3 also translocate to the tips of tubules?

Response #12

Unfortunately, because the specific antibodies to Met (CST, clone 25H2, #3127), STAT3 (CST, #9132), and STAT3 pY705 (CST, #9131) were not applicable for immunofluorescence microscopic analyses of three-dimensional culture of MDCK cells (see Fig. 3 for review), we could not visualize Met and STAT3 in our experimental system. However, DRM separation analysis showed that HGF stimulation induced Met translocation (**Fig. S1D**) and overexpressed CDCP1 recruited Met into the lipid raft-enriched fractions (**Fig. S8D**).

8. EGFR is known to interact with CDCP1 - is it clear that EGFR is not also involved in HGF-induced formation of protrusions?

Response #13

Following the reviewer's suggestion, we examined the contribution of EGFR to the HGF-induced responses. We found that the HGF-induced protrusion formation was not suppressed by pretreatment with high concentration (500 nM) of EGFR specific inhibitors lapatinib (IC₅₀, 10.8 nM), gefitinib (IC₅₀, 20-50 nM) or erlotinib (IC₅₀, 2 nM) (**Fig. S13**). Furthermore, the CDCP1-mediated protrusion formation and multilayer formation were not suppressed by the pretreatment with these inhibitors (**Table S1**). These data indicate that HGF- or CDCP1-mediated protrusion formation does not require EGFR kinase activity. We included these

data in the revised version of the manuscript.

There are also some other issues:

1. Dasatinib is a very "dirty" kinase inhibitor, inhibiting many Tyr (and Ser/Thr) kinases, including the collagen-activated DDR1/2 RTKs (since the MDCK cysts are cultured in collagen, this could be an issue). Moreover, while dasatinib does inhibit c-Src. it inhibits the Yes and Fyn SFKs equally well, and also inhibits Abl (and Arg/Abl2) just as potently. This is important, because Abl is known to induce formation of cellular protrusions when activated (e.g. Woodring et al., JCB 156:879, 2002). Thus, the use of dasatinib as a Src inhibitor is not advised, since any observed phenotypes cannot necessarily be attributed to SFK inhibition. Significantly more selective SFK inhibitors, such as saracatinib, are available, If the authors want to establish a role for Src per se, they should generate Src^{-/-} MDCK cells.

Response #14

In accordance with the comments, we analyzed MDCK cells in the presence of saracatinib to ensure the contribution of other SFK proteins. Please refer to Response #1.

2. Figure S1A: The authors need to remind the readers of the structure of the Src-MER protein, which they developed previously, i.e. the MER domain is appended to the C-terminus, and indicate whether the presence of the C-terminal MER domain affects negative regulation of Src activity by Csk phosphorylation.

Response #15

Following the reviewer's comment, we added a schematic diagram and legend of Src-MER protein (**Fig. S2A**) and rewritten the material and method.

3. Figure 2C: It appears that there is residual CDCP1 in the CDCP1KO#1 lane - does this mean that these Cdcp1^{-/-} MDCK cells were not clonal?

Response #16

CDCP1 KO cells used in this study are clonal. When we checked the CDCP1 knockout clone cells by western blot using an antibody against canis CDCP1 (LSBio, clone 5B3, LS-C172540), some non-specific bands were detected in long-exposure images (see Fig. 4 for review).

[Figure removed by editorial staff per authors' request]

4. Is there a role for any of the other pTyr sites in CDCP1 in this pathway, and is their phosphorylation induced by HGF treatment?

Response #17

CDCP1 protein has multiple tyrosine phosphorylation sites. Phosphorylation of Tyr762 provides a binding site of PKC δ and activates downstream pathways leading to cell migration, growth, and survival (Uekita, *Mol Cell Biol*, 2007; Nakashima, *Cancer Sci*, 2017). We introduced a point mutation at Tyr762 (Y762F) and overexpressed in MDCK cysts. Overexpression of CDCP1-Y762F-EGFP induced formation of protrusions and multi-layered structures (**Fig. S12C**), similarly to wild-type CDCP1-EGFP-overexpressed cysts. Furthermore, pretreatment with 400 nM PKC inhibitor, Gö6983 (IC₅₀, 10 nM for PKC δ), did not suppress CDCP1-mediated phenotypic changes (**Fig. S12D**). Taken together, phosphorylation of Tyr762 was not involved in CDCP1-mediated morphological changes in our experimental models.

Reviewer #2 (Comments to the Authors (Required)):

In this study by Kajiwara, et al entitled "CDCP1 promotes compensatory renal growth by integrating Src and Met signalling" the authors present a novel signalling axis involving the scaffolding protein CDCP1, HGF and STAT3 signaling, which controls epithelial cell proliferation and ECM remodelling. Moreover, the authors integrate this signalling network in the context of compensatory organ growth and employ unilateral nephrectomy in the mouse as a model system.

In general, the study is employing a wide variety of different methodologies including complex 3D culture systems, genome engineering, proteomics and biochemistry approaches as well as in vivo mouse models. The majority of experiments are performed in a very stringent and detailed fashion, accompanied by a clear and easy-to-follow description in the manuscript.

The manuscript is clearly separated into two different aspects, where one is exclusively using MDCK cells in 3D as a model system for renal tubular epithelial cells and the other one is analysing the Cdcpl mouse model. Here, each part for its own is very elegantly presented (the in vitro part in much more detail), but the synthesis, combination and conclusions from both parts are less convincing than the individual aspects. Based on that the authors should be careful in the interpretation of the data and conclusions.

1) MDCK system and in vitro analysis:

This part of the manuscript is really extensive and provides interesting novel insights about Cdcpl function in the MDCK cyst model. Genetic as well as inhibitor studies are elegantly performed.

Following points should be addressed:

-the authors use a wide variety of chemical compounds and inhibitors (e.g. figure 1) - here they state that mTOR inhibition suppresses cell proliferation of cysts. The presented IF images also illustrate a dramatically altered morphology of individual cysts. The authors should also discuss effects on the cytoskeleton or viability of the cells due to compound treatments.

Response #18

In accordance with the reviewer's comment, we discussed cytoskeletal disorganization of MDCK cysts in the presence of mTOR inhibitor (**Fig. 1A**) in the Discussion section.

Also the statement that dasatinib suppresses proliferation is definitely true looking at the cyst diameters, however at the same time Ki67 positivity is rather unaffected in the presented images. Maybe Ki67 quantification of cysts could clarify the impact on proliferation and could exclude secondary effects on cyst morphology due to cytoskeleton rearrangement or increased cell death (or cell volume regulation?).

Response #19

In accordance with the reviewer's comment, we measured the ratio of Ki67 positive cells in MDCK cysts in the presence of inhibitors and presented new graphs (**Fig. 1C, 3G S1B, S6E, and S13B**).

-the authors argue that CDCP1 is tethered to lipid rafts (this is also based on biochemistry experiments). In Figure 1 the authors use mCherry-GPI sensors to demonstrate the localization of CDCP1 towards lipid rafts, but the IF stainings show more or less exclusive localization at cellular protrusions (would a co-staining approach be feasible to demonstrate a clear co-localization for the protein and the sensor)? This would help to clarify the exact localization pattern of CDCP1 in 3D culture.

Response #20

Thank you for your valuable comment. We reanalyzed the localization of mCherry-GPI and other proteins in three-dimensional culture several times. However, we could not obtain good images using our fluorescence microscopy system. Instead of immunostaining analysis, we performed biochemical analysis using a lipid raft (DRM) isolation technique and comprehensively evaluated protein translocation into lipid rafts (**Fig. S1D**). Time-course analysis revealed that SFKs were present in the DRM fraction and were gradually phosphorylated following HGF treatment. CDCP1 was also constitutively detected in the DRM fraction, and its phosphorylation was observed after SFK activation. Met was translocated into the DRM fraction, and phosphorylated Met was detected in the DRM fraction at the same time as CDCP1 phosphorylation. These biochemical analyses provided new evidence for the dynamics of protein phosphorylation inside lipid rafts.

-based on the cellular phenotype and also the microarray data the authors identify ECM remodelling as a central downstream effect of CDCP1 function (also involving MMP signalling). This is also elegantly investigated using genetic interference and chemical compounds. In the discussion the authors mention that CDCP1 is also promoting this kind of phenotype in cancer tissues, however, it is less clear how this "invasive" phenotype might reflect any aspect of kidney tubule function? ECM degradation (as required for this invasive behaviour) is not required for tubular regeneration or growth. Based on that, it is highly questionable whether this phenotype really reflects kidney epithelial cell behaviour in a meaningful way. The reviewer is well aware that this model (incl. HGF application) is widely used and accepted, but nevertheless the authors should clearly state that this is only used as a proxy to model CDCP1 effects in epithelial cells (see also below). The transfer of the observed phenotype to the *in vivo* condition is clearly a difficult point in the whole manuscript.

Response #21

We agree with the reviewer's comment that there are differences between invasive-like phenotype of MDCK cells and the behavior of renal epithelial cells. In the experiments that used MDCK cysts as *in vitro* models,

CDCP1 was forcibly and stably overexpressed, and CDCP1-induced phenomena were clearly enhanced and provided convincing evidence. These *in vitro* experiments were useful for analyzing the molecular details of CDCP1 functions. Although the role of CDCP1-Src in tubular extension, that is, the invasive-like phenotype, was not verified *in vivo*, likely because of the lower expression level of endogenous CdcP1 in the kidney, some effects of CdcP1 KO on HGF-induced transient and focal phenomena, such as the transient activation of Met, Src, STAT3, and MMPs, were successfully appreciated. Thus, we believe that the *in vivo* observations together with the *in vitro* data underscore the crucial role of CDCP1-Src at least in the regulation of the HGF-Met-STAT3 pathway. However, we understand that the role of CDCP1-induced invasive phenotypes in renal regeneration must be addressed more carefully. We have added these statements in the Discussion section and improved the manuscript.

2) CdcP1 mouse model:

-the tubular thickening phenotype described in the UNX model is really questionable - at least on the presented images the difference between WT and KO mice is not easily to distinguish. Has this kind of quantification been used before in other publications? Is this approach validated at different time points? Such kind of measurements are really prone for artefacts and therefore a clear description of sample preparation and measurement and quantification approach is required.

Response #22

Based on the reviewer's suggestion, we reassessed the changes in proximal tubules based on a previous report (Chen, *J Clin Invest*, 2015; Liu, *Am J Physiol Renal Physiol*, 2020). The area of the proximal tubule was measured using immunofluorescence images of kidney slices visualized by LTL- (apical side) and collagen IV-staining (basal side) (**Fig. 6D and F**). Furthermore, the area of glomeruli was measured using HE staining of kidney slices (**Fig. 6C and G**). All measurements were performed in 50 tubules/glomeruli of five mice/group and subjected to ANOVA statistical analysis.

In addition, the HE slides of the kidney from the experimental animal were reviewed by certified pathologist (Diplomate of Japanese Society of Toxicologic Pathology) who evaluated them and provided a diagnosis of the pathological findings, including inflammation, degeneration, and necrosis. Consequently, the above-mentioned remarkable findings were not observed in all animals, although slight changes including glomerular and renal tubule hypertrophy were observed in the UNX model.

Also HE/PAS stains of the kidneys should be presented, preferably whole slide scans and close ups.

Response #23

In accordance with the reviewer's suggestion, we presented detailed and whole HE staining data of kidney

sections (**Fig. 6C and Fig. S10D**, respectively).

-what is the absolute effect on the kidney weight? Only kidney/body weight ratios are presented making it difficult to grasp the absolute effect on the kidney mass. While the difference in ratios between WT/KO mice is significant, the delta of the ratio is rather modest. This makes it a little bit questionable whether knockout mice really have a dramatic deficit in organ growth. These points should be addressed by the authors (this holds also true for the proliferation rate - here is only a difference of about 2% observed in the Ki67 proliferation rate; whether that might really relate to impaired organ growth needs to be discussed).

Response #24

In accordance with the reviewer's comment, we have presented the absolute effects on kidney weight in WT and KO mice (**Fig. 7B and Fig. S10A–C**). We divided “compensatory growth” into “hypertrophy and hyperplasia” and explained our results using this definition. According to a previous study (Johnson and Roman, *Am J Pathol*, 1966), cell hypertrophy is an immediate (< 2 days) and predominant response (three-fourth of increase), whereas hyperplasia is a post-immediate response (> 2 days). In addition to Ki67-positive proliferation rate, in KO mice, immediate hypertrophy was weaker than that in the WT mice. These results suggest that *Cdcp1* is required for the efficient progression of both hypertrophy and hyperplasia.

-is there any other published evidence of collagen IV breakdown in the tubular basement membrane (TBM) in conditions of UNX or tubular damage? The TBM is usually a very stable structure and dissolution is only observed in degenerating tubular compartments. As the authors try to connect the ECM/invasion phenotype from their in vitro model with the in vivo situation, the authors should (if they really want to emphasize this point) try to quantitate these effects and corroborate those findings by ultrastructural techniques. Based on published work this aspect of organ growth and TBM remodelling is not really established.

Response #25

In accordance with the reviewer's comment, we quantified immunofluorescence staining data for collagen IV and performed statistical analysis (**Fig. S11E**). We observed breakdown of the basement membrane around the proximal tubules 2 days after UNX in some parts of the proximal tubules (**Fig. S11D and E**). After 2 days, the basement membrane recovered, and differences between the sham-operated and nephrectomized mice became undetectable, except for upregulation of collagen IV staining (**Fig. 6D**). In addition, secretion of MMP2/9 was visualized 1 day after nephrectomy (**Fig. S11F and G**). Previous studies have reported that expression of ECM- and MMP-related genes is transiently upregulated after UNX (Nakamura, *AJP Renal Physiol*, 1992; Koide, *Nephron*, 1997; Zhang, *Kidney Int*, 1999) and that MMP9 expression is increased in patients with compensatory renal growth (Yildiz, *Clin Physiol Funct Imaging*, 2008). Collectively, partial degradation of

collagen IV matrix was a transient and local change, and our findings may be the first report of a transient alteration of basement membrane integrity during compensatory renal growth.

Also the cited reference "34" has a complete different focus.

Response #26

In accordance with the reviewer's comment, reference #34 was removed.

-minor: IF stainings should be quantified, where differences have been observed.

Response #27

In accordance with the reviewer's comment, we requantified immunofluorescence staining data and performed statistical analysis (**Fig. 2E, 3E, 5C, S6J, S11E, and S12F**).

Reviewer #3 (Comments to the Authors (Required)):

Kajiwara et al. identify a function of CDCP1 in regulating changes in morphology and growth of renal cysts. CDCP1 expression is required and sufficient for formation of protrusions in MDCK cell cysts; moreover, CDCP1 deficiency prevents renal cyst growth in an in vivo mouse model. The authors confirm a role for CDCP1 in Src - STAT3 signaling to mediate these changes and provide evidence that these signaling events require lipid raft localization of CDCP1-Src. The present study extends known functions of CDCP1 in signal transduction by interesting observations about the importance of subcellular localization and the implication for renal growth. The experiments are of overall high quality and support the author's conclusions. The data are clearly presented in a well-written manuscript, and I recommend this work for publication in Life Science Alliance. A few suggestions to further improve the manuscript are outlined below.

1. The authors propose in their model (Fig. 6H) that CDCP1-Src specifically upregulate Stat3 downstream of Met, suggesting the interesting possibility that CDCP1 fine-tunes Met signaling by selective activation of some downstream effectors. Consistently, I have the impression that Dasatinib potently suppresses formation of cyst protrusions but is less effective than Torin 1 or Rapamycin in suppressing cell proliferation (Fig. 1A-C). It would be interesting to know whether CDCP1 overexpression / inhibition influences mTORC1 activity in HGF-stimulated cells.

Response #28

Thank you for the interesting comment. We are also interested in CDCP1-dependent regulation of the mTORC1 pathway. We observed that overexpression of CDCP1 induced Met phosphorylation and activated multiple downstream pathways, including MAPK and mTOR signaling (data not shown). We are currently analyzing this regulation and intend to report it separately in the near future.

2. Fig. 1D, F: If possible, it would be good to co-stain for p-Src and CDCP1 in the same sample.

Response #29

Please refer response #20.

3. Suppl. Fig. 1C: Do the numbers on the left side indicate time? If yes, which unit?

Response #30

These numbers indicate time (hour) after treatment with 4-OHT (Fig. S2D). We improved description of figure and its legend.

4. Suppl. Fig. 7B: Which biological samples were used for the PCR? A Western blot of CDPC1 in wt / KO would be helpful.

Response #31

We used a piece of mouse ear as a sample for PCR (**Fig. S9C**). In accordance with the reviewer's comment, we newly presented western blot data of kidney samples from wild-type, Cdcp1^{+/-} and Cdcp1^{-/-} mice (**Fig. S9D**).

5. Suppl. Fig. 8A: It is a bit difficult to tell which p-CDPC1 staining is specific vs non-specific (compare e.g. Sham Cdcp1^{-/-} and UNX 0.5 d Cdcp1^{+/+}).

Response #32

In accordance with the reviewer's comment, we replaced immunofluorescence staining data (**Fig. 7E**).

6. In the methods section, please give the antibody incubation times for immunoprecipitation and immunostainings.

Response #33

We apologize for the poorly written method section. We rewrote the method of immunoprecipitation and immunostaining analyses.

January 12, 2021

RE: Life Science Alliance Manuscript #LSA-2020-00832-TR

Dr. Kentaro Kajiwara
Research Institute for Microbial Diseases, Osaka University
Department of Oncogene Research
Yamadaoka 3-1
Suita, Osaka 565-0871
Japan

Dear Dr. Kajiwara,

Thank you for submitting your revised manuscript entitled "CDCP1 promotes compensatory renal growth by integrating Src and Met signaling". We would be happy to publish your paper in Life Science Alliance pending final revisions necessary to meet the reviewers' requests and our formatting guidelines.

Along with the points listed below, please attend to the following,

-please use the [10 author names, et al.] format in your references (i.e. limit the author names to the first 10)

-please address the requests raised by Reviewer 2 through textual and figure schematic edits.
Please provide a pbp rebuttal to the reviewers' comments

A. FINAL FILES:

-- Summary blurb (enter in submission system): A short text summarizing in a single sentence the study (max. 200 characters including spaces). This text is used in conjunction with the titles of

papers, hence should be informative and complementary to the title. It should describe the context and significance of the findings for a general readership; it should be written in the present tense and refer to the work in the third person. Author names should not be mentioned.

B. MANUSCRIPT ORGANIZATION AND FORMATTING:

Sincerely,

Shachi Bhatt, Ph.D.
Executive Editor
Life Science Alliance
<https://www.lsjournal.org/>
Tweet @SciBhatt @LSAJournal

Reviewer #1 (Comments to the Authors (Required)):

In the revised version, the authors have done an excellent job of addressing the many issues that I

raised, by adding a large amount of new data, including the new Figure 5, which provides more mechanistic insights into how CDCP1 focally integrates local Src and Met-STAT3 signaling in lipid rafts. They have also added new experiments with the specific SFK inhibitor saracatinib, which support their conclusion that Src family kinases play an important role in HGF/Met signaling and also provide evidence that the Yes SFK may play a role in addition to Src. Unfortunately, Figures 6 and 7 were omitted from the compiled pdf of the revised version that I downloaded (and they are not available as single files either), but according to the rebuttal the authors' reassessment of tubule thickening and hypertrophy in the remaining kidney after UNX led them to conclude that the decrease in thickening in *Cdcp1*^{-/-} was not evident in all mice, whereas a decrease in immediate hypertrophy was consistent. It is good this was the authors were able to correct their previous interpretations of the in vivo experiments, and reassuring that there were significant differences in compensatory renal growth in *Cdcp1*^{-/-} mice. This paper is suitable for publication without further revision.

Reviewer #2 (Comments to the Authors (Required)):

In this revised version of the manuscript the authors have largely addressed the issues raised in the initial review process (I've summarized the initial report, and based on the response of the authors the actual evaluation). Except for little adjustments in the text (e.g. discussing the problem of transfer from in vitro to in vivo observations, or adding references) there is left only one issue with the aspect of ECM rearrangement. This is largely based on the observation that STAT-signalling affects the expression values of MMP9 (in vitro). The validation/correlation for the in vivo situation is solely only based on IF-staining (where difficulties in staining/visualizing MMP proteins are well known). While the compensatory kidney growth phenotype based on histological/IF evaluations appear overall sound in the context of the extensive and very detailed in vitro experiments, the authors should be careful in overemphasizing the mechanistic and underlying concept in vivo. Taking into account the presented data, one should probably interpret the observed changes (ColIV staining intensities) more carefully and at least adapt the presented schematics accordingly (e.g. dashed lines to indicate potential impact in the in vivo situation).

Individual points raised in the initial revision:

1) the authors use a wide variety of chemical compounds and inhibitors (e.g. figure 1) -here they state that mTOR inhibition suppresses cell proliferation of cysts. The presented IF images also illustrate a dramatically altered morphology of individual cysts. The authors should also discuss effects on the cytoskeleton or viability of the cells due to compound treatments.

R: The authors adequately addressed this point in the revised discussion section.

2) Also the statement that dasatinib suppresses proliferation is definitely true looking at the cyst diameters, however at the same time Ki67 positivity is rather unaffected in the presented images. Maybe Ki67 quantification of cysts could clarify the impact on proliferation and could exclude secondary effects on cyst morphology due to cytoskeleton rearrangement or increased cell death (or cell volume regulation?).

R: The authors adequately addressed this point in the revised version of the manuscript.

3) the authors argue that CDCP1 is tethered to lipid rafts (this is also based on biochemistry experiments). In Figure 1 the authors use mCherry-GPI sensors to demonstrate the localization of

CDCP1 towards lipid rafts, but the IF stainings show more or less exclusive localization at cellular protrusions (would a co-staining approach be feasible to demonstrate a clear co-localization for the protein and the sensor)? This would help to clarify the exact localization pattern of CDCP1 in 3D culture.

R: The novel biochemical isolation/fractionation experiments are supporting the initial descriptions.

-based on the cellular phenotype and also the microarray data the authors identify ECM remodelling as a central downstream effect of CDCP1 function (also involving MMP signalling). This is also elegantly investigated using genetic interference and chemical compounds. In the discussion the authors mention that CDCP1 is also promoting this kind of phenotype in cancer tissues, however, it is less clear how this "invasive" phenotype might reflect any aspect of kidney tubule function? ECM degradation (as required for this invasive behaviour) is not required for tubular regeneration or growth. Based on that, it is highly questionable whether this phenotype really reflects kidney epithelial cell behaviour in a meaningful way. The reviewer is well aware that this model (incl. HGF application) is widely used and accepted, but nevertheless the authors should clearly state that this is only used as a proxy to model CDCP1 effects in epithelial cells (see also below). The transfer of the observed phenotype to the in vivo condition is clearly a difficult point in the whole manuscript.

R: The reviewer appreciates the changes in the revised version of the manuscript. However, a clearer description that the observed invasive phenotype (MDCK) was used as model to study CDCP1 effects should be integrated - translation of the in vitro findings to the in vivo observations is still difficult.

2) CdcP1 mouse model: -the tubular thickening phenotype described in the UNX model is really questionable -at least on the presented images the difference between WT and KO mice is not easily to distinguish. Has this kind of quantification been used before in other publications? Is this approach validated at different time points? Such kind of measurements are really prone for artefacts and therefore a clear description of sample preparation and measurement and quantification approach is required.

R: The reviewer appreciates the effort to quantify tubule thickness as well as tubular area derived from IF and histological stainings. The mentioned references for the methodological approach should be included.

-what is the absolute effect on the kidney weight? Only kidney/body weight ratios are presented making it difficult to grasp the absolute effect on the kidney mass. While the difference in ratios between WT/KO mice is significant, the delta of the ratio is rather modest. This makes it a little bit questionable whether knockout mice really have a dramatic deficit in organ growth. These points should be addressed by the authors (this holds also true for the proliferation rate -here is only a difference of about 2% observed in the Ki67 proliferation rate; whether that might really relate to impaired organ growth needs to be discussed).

R: The additional explanation to describe the adaptive response (hypertrophy and hyperplasia) is adding to the overall concept of the paper.

-is there any other published evidence of collagen IV breakdown in the tubular basement membrane (TBM) in conditions of UNX or tubular damage? The TBM is usually a very stable structure and dissolution is only observed in degenerating tubular compartments. As the authors try

to connect the ECM/invasion phenotype from their in vitro model with the in vivo situation, the authors should (if they really want to emphasize this point) try to quantitate these effects and corroborate those findings by ultrastructural techniques. Based on published work this aspect of organ growth and TBM remodelling is not really established.

R: Thanks for the explanation. However, in results shown in Figure S11D and E do not match. If the WT mice would show a higher ColIV expression at later time points, the images in S11E are probably mixed up? How was the IF-measurement for CollIV performed? The IF-intensities are in some images globally decreases (also LTL-Staining) - this could affect the measurement. The study cited (Yildiz, 2008) measured serum concentrations and indeed found elevated levels for several growth factors as well as MMP9. However, this is not really relating to any structural changes in tubular basement membranes, but reflects only global changes in MMP9 serum concentrations. If the authors really want to strengthen these observations, one could imagine that isolation of ECM fractions of kidneys could be used for western blot evaluation or evaluation of basement membranes using electron microscopy.

Reviewer #3 (Comments to the Authors (Required)):

The authors have addressed my concerns to my satisfaction and I recommend the paper for publication in Life Science Alliance.

Point-by-point response letter

LSA-2020-00832-TR

CDCP1 promotes compensatory renal growth by integrating Src and Met signaling

Kentarō Kajiwara, Shotaro Yamano, Kazuhiro Aoki, Daisuke Okuzaki, Kunio Matsumoto and Masato Okada

Reviewer #2 (Comments to the Authors (Required)):

In this revised version of the manuscript the authors have largely addressed the issues raised in the initial review process (I've summarized the initial report, and based on the response of the authors the actual evaluation). Except for little adjustments in the text (e.g. discussing the problem of transfer from in vitro to in vivo observations, or adding references) there is left only one issue with the aspect of ECM rearrangement. This is largely based on the observation that STAT-signalling affects the expression values of MMP9 (in vitro). The validation/correlation for the in vivo situation is solely only based on IF-staining (where difficulties in staining/visualizing MMP proteins are well known). While the compensatory kidney growth phenotype based on histological/IF evaluations appear overall sound in the context of the extensive and very detailed in vitro experiments, the authors should be careful in overemphasizing the mechanistic and underlying concept in vivo. Taking into account the presented data, one should probably interpret the observed changes (ColIV staining intensities) more carefully and at least adapt the presented schematics accordingly (e.g. dashed lines to indicate potential impact in the in vivo situation).

Response #34

Thank you for your comments and practical suggestions. We understood your considerations and modified the schematic model in Figure 7H.

Individual points raised in the initial revision:

1) the authors use a wide variety of chemical compounds and inhibitors (e.g. figure 1) -here they state that mTOR inhibition suppresses cell proliferation of cysts. The presented IF images also illustrate a dramatically altered morphology of individual cysts. The authors should also discuss effects on the cytoskeleton or viability of the cells due to compound treatments.

R: The authors adequately addressed this point in the revised discussion section.

2) Also the statement that dasatinib suppresses proliferation is definitely true looking at the cyst diameters, however at the same time Ki67 positivity is rather unaffected in the presented images. Maybe Ki67 quantification of cysts could clarify the impact on proliferation and could exclude secondary effects on cyst morphology due to cytoskeleton rearrangement or increased cell death (or cell volume regulation?).

R: The authors adequately addressed this point in the revised version of the manuscript.

3) the authors argue that CDCP1 is tethered to lipid rafts (this is also based on biochemistry experiments). In Figure 1 the authors use mCherry-GPI sensors to demonstrate the localization of CDCP1 towards lipid rafts, but the IF stainings show more or less exclusive localization at cellular protrusions (would a co-staining approach be feasible to demonstrate a clear co-localization for the protein and the sensor)? This would help to clarify the exact localization pattern of CDCP1 in 3D culture.

R: The novel biochemical isolation/fractionation experiments are supporting the initial descriptions.

-based on the cellular phenotype and also the microarray data the authors identify ECM remodelling as a central downstream effect of CDCP1 function (also involving MMP signalling). This is also elegantly investigated using genetic interference and chemical compounds. In the discussion the authors mention that CDCP1 is also promoting this kind of phenotype in cancer tissues, however, it is less clear how this "invasive" phenotype might reflect any aspect of kidney tubule function? ECM degradation (as required for this invasive behaviour) is not required for tubular regeneration or growth. Based on that, it is highly questionable whether this phenotype really reflects kidney epithelial cell behaviour in a meaningful way. The reviewer is well aware that this model (incl. HGF application) is widely used and accepted, but nevertheless the authors should clearly state that this is only used as a proxy to model CDCP1 effects in epithelial cells (see also below). The transfer of the observed phenotype to the in vivo condition is clearly a difficult point in the whole manuscript.

R: The reviewer appreciates the changes in the revised version of the manuscript. However, a clearer description that the observed invasive phenotype (MDCK) was used as model to study CDCP1 effects should be integrated - translation of the in vitro findings to the in vivo observations is still difficult.

Response #35

In accordance with the reviewer's comment, we further modified the manuscript, especially in the discussion section. Additionally, we removed the confusing keyword "invasive growth" from the result section of the revised manuscript.

2) Cdcp1 mouse model: -the tubular thickening phenotype described in the UNX model is really questionable -at least on the presented images the difference between WT and KO mice is not easily to distinguish. Has this kind of quantification been used before in other publications? Is this approach validated at different time points? Such kind of measurements are really prone for artefacts and therefore a clear description of sample preparation and measurement and quantification approach is required.

R: The reviewer appreciates the effort to quantify tubule thickness as well as tubular area derived from IF and histological stainings. The mentioned references for the methodological approach should be included.

Response #36

We analyzed tubular thickness by using the methodological approach based on the previous study (Endele, *Transgenic Res*, 2007) and added this reference to the new version of the manuscript.

-what is the absolute effect on the kidney weight? Only kidney/body weight ratios are presented making it difficult to grasp the absolute effect on the kidney mass. While the difference in ratios between WT/KO mice is significant, the delta of the ratio is rather modest. This makes it a little bit questionable whether knockout mice really have a dramatic deficit in organ growth. These points should be addressed by the authors (this holds also true for the proliferation rate -here is only a difference of about 2% observed in the Ki67 proliferation rate; whether that might really relate to impaired organ growth needs to be discussed).

R: The additional explanation to describe the adaptive response (hypertrophy and hyperplasia) is adding to the overall concept of the paper.

-is there any other published evidence of collagen IV breakdown in the tubular basement membrane (TBM) in conditions of UNX or tubular damage? The TBM is usually a very stable structure and dissolution is only observed in degenerating tubular compartments. As the authors try to connect the ECM/invasion phenotype from their in vitro model with the in vivo situation, the authors should (if they really want to emphasize this point) try to quantitate these effects and corroborate those findings by ultrastructural techniques. Based on published work this aspect of organ growth and TBM remodelling is not really established.

R: Thanks for the explanation. However, in results shown in Figure S11D and E do not match. If the WT mice would show a higher ColIV expression at later time points, the images in S11E are probably mixed up? How was the IF-measurement for ColIV performed? The IF-intensities are in some images globally decreases (also LTL-Staining) - this could affect the measurement. The study cited (Yildiz, 2008) measured serum

concentrations and indeed found elevated levels for several growth factors as well as MMP9. However, this is not really relating to any structural changes in tubular basement membranes, but reflects only global changes in MMP9 serum concentrations. If the authors really want to strengthen these observations, one could imagine that isolation of ECM fractions of kidneys could be used for western blot evaluation or evaluation of basement membranes using electron microscopy.

Response #37

Thank you for your comment and valuable suggestion. We think these provided points are important and valuable for further understanding the compensatory renal growth. We would like to further analyze the alterations of ECM environment using various methods including electron microscopic observations and report it separately in the future.

Fluorescence intensity of collagen IV surrounding proximal tubules was carefully measured in a non-biased manner. We randomly chose proximal tubules with comparable fluorescence intensity of LTL-FITC and analyzed 50 circular tubules (5 mice/group) using the ImageJ software. Following the reviewer's comments, to avoid confusing interpretations of the results, we changed the immunofluorescence image in Figure S11D. The images in Figure S11D and the graph in Figure S11E are correct, and Figure S11E were created from data presented in Figure S11D (UNX 2 days) and Figure 6D (UNX 56 days).

January 19, 2021

RE: Life Science Alliance Manuscript #LSA-2020-00832-TRR

Dr. Kentaro Kajiwara
Research Institute for Microbial Diseases, Osaka University
Department of Oncogene Research
Yamadaoka 3-1
Suita, Osaka 565-0871
Japan

Dear Dr. Kajiwara,

Thank you for submitting your Research Article entitled "CDCP1 promotes compensatory renal growth by integrating Src and Met signaling". It is a pleasure to let you know that your manuscript is now accepted for publication in Life Science Alliance. Congratulations on this interesting work.

DISTRIBUTION OF MATERIALS:

Again, congratulations on a very nice paper. I hope you found the review process to be constructive and are pleased with how the manuscript was handled editorially. We look forward to future exciting submissions from your lab.

Sincerely,

Shachi Bhatt, Ph.D.

Executive Editor

Life Science Alliance

<https://www.lsjournal.org/>
